# Environmental Effects on Taxonomic Turnover in Soil Fauna across Multiple Forest Ecosystems in East Asia

**DOI:** 10.3390/insects13121103

**Published:** 2022-11-30

**Authors:** Peikun Li, Jian Zhang, Shunping Ding, Peisen Yan, Panpan Zhang, Shengyan Ding

**Affiliations:** 1Key Laboratory of Geospatial Technology for the Middle and Lower Yellow River Regions, Ministry of Education, College of Geography and Environmental Science, Henan University, Kaifeng 475004, China; 2College of Geography and Environmental Science, Henan University, Kaifeng 475004, China; 3Plant Sciences Department, California Polytechnic State University, San Luis Obispo, CA 93407, USA

**Keywords:** community structure, East Asia, environmental filtering, latitudinal gradient, soil meso- and macrofauna

## Abstract

**Simple Summary:**

The mechanisms determining biotic turnover across geographic gradients are poorly understood. We obtained the catalogues of soil fauna orders across forest ecosystems in East Asia as well as the temperature, moisture and soil physical and chemical properties of the corresponding sites. We then studied the composition of soil fauna orders in different climatic regions and analyzed the effect of environmental factors on the composition of the soil fauna community in East Asia. Finally, we calculated the total variation in environmental and spatial factors in the orders’ composition of overall, phytophagous, predatory and saprophagous fauna and explored the large-scale patterns of soil meso-macrofauna turnover and the driving factors based on fourteen sampling sites in East Asia. The results showed that the patterns of soil fauna order turnover increased significantly with increasing latitude differences. The total variance in the environment was higher than the spatial factors in the orders’ composition. Meanwhile, the effects of climate factors on environmental processes were stronger than those of soil factors on phytophage, predacity and saprophage faunas. The results help us better understand, maintain and promote soil biodiversity conservation and ecosystem services at large scales.

**Abstract:**

The large-scale spatial variation in and causes of biotic turnover of soil fauna remain poorly understood. Analyses were conducted based on published data from 14 independent sampling sites across five forest ecosystems in East Asia. Jaccard and Sørensen’s indices were used to measure turnover rates in soil fauna orders. A redundancy analysis was used to investigate multiple environmental controls of the composition of soil fauna communities. The results showed that both Jaccard’s and Sørensen’s index increased significantly with increasing latitude difference. The environment explained 54.1%, 50.6%, 57.3% and 50.9% of the total variance, and spatial factors explained 13.8%, 15.9%, 21.0% and 12.6% of the total variance in the orders’ composition regarding overall, phytophagous, predatory and saprophagous fauna, respectively. In addition, climate factors in environmental processes were observed to have a stronger effect than soil factors on the orders’ turnover rates. Our results support the hypothesis that the effect of environment factors on soil animal taxa turnover is more important than the effect of spatial factors. Climatic factors explained more variation in the turnover of phytophagic fauna, but soil and environment factors equally explained the variation in the turnover of predatory fauna. This study provides evidence to support both environmental filtering and dispersal limitation hypotheses at the regional and population scales.

## 1. Introduction

Soils are the most biodiverse habitats on Earth, with soil fauna accounting for about 23% of all described biodiversity globally [1]. Based on body length, they are usually divided into microfauna (average length of less than 0.1 or 0.2 mm, such as Protozoa and nematodes), mesofauna (average length between 0.2 and 2 mm, such as Collembola and Acari), macrofauna (average length larger than 2 mm, such as earthworms and spiders), and mega-fauna (average length larger than 2 cm, such as moles) [2]. Soil fauna primarily contribute to ecosystem functioning through trophic and/or non-trophic effects [1]. Most micro- and/or meso-faunal groups can regulate soil microbial processes directly by feeding on detritus and microbes, which indirectly affects soil structure and soil carbon and nitrogen processes [3]. In addition, predators such as spiders can affect soil ecosystems via the trophic cascade effects of predators [4]. However, macrofauna, such as earthworms, may regulate soil microbes and thus alter soil carbon and nitrogen processes considerably by non-trophic effects [5]. In addition, there are some phytophagous, saprozoic and omnivorous soil faunas, which are often active on the soil surface and have high biotic richness and diversity; such fauna also have an important impact on the soil ecosystem [1]. The diverse feeding strategies of soil fauna and the complex non-trophic relationships establish multi-dimensional soil food webs, which play a crucial role in the material cycle and energy flow of terrestrial ecosystems [6]. Moreover, soil animal communities may also alter litter decomposition, nutrient mineralization, soil respiration and plant community composition [7,8,9,10]. Consequently, shifts in soil animal community composition could dramatically influence the functioning and stability of terrestrial ecosystems [11]. However, little is known about the spatial and environmental factors that shape soil animal communities on a regional and global scale [12]. To understand ecosystem functioning and the mechanisms of community composition evolvement, it is necessary to identify the factors that shape the distribution and structure of various soil faunas [13]. However, collecting and recording biotic diversity information is challenging, especially in soil fauna [14]. In recent decades, in China, ecologists have widely used the order taxon to study the diversity and ecogeographic distribution of soil fauna communities in different temperature zones and ecosystems and achieved remarkable results [15,16].

Diversity indexes are often used to evaluate the status of communities or ecosystems and usually include α-diversity, which indicates diversity within the same community, and β-diversity, which indicates biodiversity among different communities [17]. Biotic turnover pattern refers to the pattern of biotic composition divergence or biotic turnover among communities in different habitats along environmental gradients [18], and it provides considerable assistance for understanding and revealing the mechanisms of community construction, especially the coexistence and community pattern of species at large scales [19,20]. However, in contrast to α-diversity, the patterns of biological turnover have been inadequately studied [21], and measures of biotic turnover in previous studies are usually based on a single indicator [22] (either Jaccard’s or Sørensen’s indices or others). However, utilizing different methods to measure biotic turnover rates can better avoid the error caused by methods of measurement. Additionally, most studies on biotic turnover have been conducted locally [23,24]. A few studies investigated biotic turnover on large geographic scales, but they mostly focused on plants, birds and mammals [25,26]. The global biogeography of soil fauna has gained attention only recently [27,28,29,30]. Most of these studies focused on specific soil fauna communities. For instance, previous studies showed that climate variability had a greater impact on earthworm community composition than soil properties [29], oribatid mites were affected by spatial factors [27], termites of the Hymenoptera order were affected by temperature and precipitation [28] and nematodes were affected by temperature [29]. Despite our accumulated knowledge about the biogeographic patterns of soil biota, the underlying mechanisms of the distribution patterns remain unexplored, especially the patterns of comprehensive taxa turnover across latitudes [31].

Environmental filtering and spatial dispersal of species are the main driving forces for changes in ecological communities and biodiversity [25]. However, their significance for the multiple dimensions of β-diversity has not been fully explored in soil fauna [32]. Several research studies indicated that biotic coexistence was attributable to differences in the biotic (e.g., community structure and species composition) and abiotic (e.g., climate and soil) environments [33,34,35]. These biotic and abiotic factors provide the habitats with different resources, time and space to achieve coexistence as implied by the niche theory [36,37]. In addition, the influence of dispersal limitation (distance effect) on community construction is one of the most important corollaries of the neutral theory, which states that biotic coexistence is due to biogeographic barriers and a low dispersal capacity [36,38]. However, these two different theories, the niche theory and neutral processes, actually jointly explain the coexistence of soil fauna communities. They have different roles on the corresponding spatial scales [39]. The underlying environmental controls that shape latitudinal shifts in soil fauna communities on a global scale, however, have not been identified [12]. The underlying mechanisms remain particularly elusive. Therefore, our study aims to address this knowledge gap and explore the influence of spatial and environmental factors on the composition of soil fauna orders in sites distributed across East Asia by synthesizing the data of soil meso- and macrofauna communities.

The objectives of the current study are: (1) to obtain an integrated analysis of the similarity of the composition of soil fauna orders along the latitude of East Asia, (2) to identify the patterns of order turnover along this latitude using two measures of taxon turnover, which include Jaccard’s and Sørensen’s indices, and (3) to assess the relative influence of environmental and neutral processes on order turnover of overall, saprophagous, phytophagous and predatory soil fauna.

## 2. Materials and Methods

### 2.1. Study Sites

In this study, we extracted data from 14 previously studied soil fauna taxa catalogs located in different climate zones across East Asia and conducted a comprehensive analysis (Figure 1). The 14 sampling locations were: Tahe, Aershan, Changbaishan, Donglingshan, Baotianman, Badagongshan, Tiantongshan, Shimentai, Jianfengling, Xishuangbanna, Guanghwa, Sapporo, Hiroshima and Yushan. The latitude of the sites range from 18.50° N to 52.33° N, and the climatic zones include temperate, subtropical and tropical zones. The vegetation types in the study areas include temperate coniferous and broad-leaved mixed forest, temperate deciduous broad-leaved forest, subtropical evergreen broad-leaved forests and tropical rain forests.

### 2.2. Taxa and Environment Variables

In the study, we used data from the taxa catalogs, which were obtained from 14 reports from the literature, as listed in Table 1 and Appendix A. The 14 sampling sites differed in established time, region and researchers (Table 1 and Table 2). The Tullgren funnel method was used for trapping soil fauna, and the soil fauna were then identified based on morphology using the same criteria in all samples [40,41]. The soil fauna identification may vary among plots at the family or order levels. Thus, the order names in the samples were checked based on the *Pictorial Keys to Soil Animals of China* [40] and *Catalogue of Life China* [42]. Concurrently, all taxa were divided into four functional types based on their feeding guilds as follows: saprozoic, omnivores, phytophagous and predatory (Table 3) [31,43].

In each sampling site, soil fauna were sampled from litter layer and/or soil cores in multiple seasons (Table 2). The sampling quantity was 339 ± 153. Additionally, the minimum individual density of soil fauna was 101.9 individuals per m^3^ in Tahe, and the maximum was 686.7 individuals per m^3^ in Jianfengling. Thus, these samplings reflect the profile of the taxa composition in that particular region well because the total sampling areas were designed to adequately cover both microhabitats and plant species [58]. Meanwhile, the 14 scientific studies selected in this paper showed that the sampling completeness of each study could fully reflect the composition of local soil fauna communities [44,45,46,47,48,49,50,51,52,53,54,55,56,57].

The longitude and latitude of the samples, climate factors (mean annual precipitation (MAP, mm), mean annual temperature (MAT, °C), mean temperature of the coldest months (MTCM, °C), extreme minimum temperature (EMT, °C)) and soil factors, including soil organic carbon (SOC, g/kg), soil bulk density (SBD, kg/m^3^) and pH, were analyzed and compared among the samples. Longitude and latitude and information on soil factors were obtained from the literature, and information on climate factors was obtained from the National Earth System Science Data Center (http://www.geodata.cn/, accessed on 1 November 2022) and Google Earth Engine (earthengine.google.com/, accessed on 1 November 2022) (Table 1 and Table 4). 

### 2.3. Measurement of Order Turnover Rate

Order turnover rate is the rate of dissimilarity among order composition across all possible plot pairs along spatial or environmental gradients. The slope of the relationship between the order turnover and environmental divergence measures order turnover rate. Jaccard’s index (βj) [59] and Sørensen’s index (βs) [60] were used to measure the turnover rate of orders’ composition. βj and βs are two widely employed indices, which only consider the presence or non-presence of orders independent of α-diversity [61]. βj and βs are calculated following the equations:βj = 1 − c/(a + b + c) = (a +b)/(a + b + c)(1)
βs = 1 − 2c/(a + b + 2c) = (a + b)/(a + b + 2c)(2)
where a and b are the numbers of orders only occurring in the focal and neighboring plots, respectively, and c is the number occurring in both.

### 2.4. Data Analysis

The 4-step iNEXT analysis, a novel class of biological survey measures, was used to quantify sample completeness and compare diversities among assemblages [62]. We then used detrended correspondence analysis (DCA) for the ordination of samples. DCA is an effective method in community analysis. In our study, we conducted DCA using a site–order matrix with relative abundance data to analyze the similarity of order composition among samples. The Kruskal–Wallis method was used to analyze differences in the richness of soil fauna in the three climatic regions.

We performed a redundancy analysis (RDA), based on variation partitioning analysis, to assess the relative effects of environmental and spatial variables on the composition of soil fauna communities. Before the RDA, the environmental variables with a high variance inflation factor (VIF) >10 were eliminated to avoid collinearity among the factors [63]. The importance of environmental and spatial variables in explaining order compositions was determined by an RDA analysis using Monte Carlo permutation tests (999 unrestricted permutations) followed by forward selection to remove the non-significant variables from each of the explanatory sets. The “envfit” function from the R package “vegan” was used to test the significance of each environmental factor for the orders’ distribution [64]. 

For environmental variables, climate and soil factors (MAP, MAT, MTCM, EMT, SOC and SBD) were used to determine the environmental divergence between pairs of sampling sites. All the environmental variables were normalized as x’ = (x - mean(x))/standard deviation (x), where x is a variable. Differences in latitude values (Table 1) of the sampling sites were used to obtain the spatial variable as a response variable. Relationships between the turnover rate of order compositions and environmental and spatial variables were determined with linear regression. We used the dissimilarity coefficient (βj and βs) as the response variable and three sets of explanatory variables, which included climate variables (MAP, MAT, MTCM and EMT), soil factors (SOC and SBD) and spatial variables (geographical co-ordinates for sampling sites). Where necessary, the values were log (x + 1) transformed in order to meet the assumptions of normality of residuals.

To further evaluate the relative importance of each environmental variable and spatial distance on the order turnover rates, we used a partial RDA (pRDA) approach. This method can analyze the effects of pivot variables and covariables on order distributions [65]. Partial RDA divided the variance in order turnover index into eight parts, which were pure spatial effects, pure climate effects, pure soil effects, spatially structured climate effects, spatially structured soil effects, climatological soil effects and the unexplained part.

All statistical analyses were carried out with R v.3.4.3 (R Core Development Team, http://www.Rproject.org, accessed on 1 November 2022). The four-step iNEXT analysis was performed using the “iNEXT4steps” package. DCA and PCA were performed using the “vegan” package [64].

## 3. Results

### 3.1. Structural Characteristics of Soil Fauna Communities

A total of 50 orders were included in the final dataset, including 14 classes in 4 phyla. In general, the number of orders declined significantly with increasing latitude for overall orders, phytophagous and predatory faunas (Figure 1). Soil fauna diversity in the temperate zone was significantly different from that in the other two climatic zones (ꭓ^2^ = 7.582, *p* = 0.02, Figure 1). The differences in phytophagous (ꭓ^2^ = 0.352, *p* = 0.839, Figure 1), predatory (ꭓ^2^ = 1.040, *p* = 0.595, Figure 1) and saprophagous (ꭓ^2^ = 1.952, *p* = 0.207, Figure 1) orders were not significant in the three climatic regions (Appendix A). Among the 14 sites studied, the Jianfengling site belongs to the tropical rain forest and showed the highest diversity of all plots (overall: 33, phytophagous: 9, predatory: 7 and saprophagous: 7). In contrast, the Aershan site, which belongs to the coniferous and broad-leaved mixed forests in the cold temperate zone, showed the lowest taxa diversity of all plots (overall: 14, phytophagous: 3, predatory: 3 and saprophagous: 2; Figure 1). 

The four-step iNEXT analysis showed that the estimated sample completeness for q = 0, 1 and 2 for the soil fauna data in East Asia was 93.1%, 97.9% and 99.6%, respectively (Figure 2, Appendix A). This means that the study data covered, at most, 93.1% of the total soil fauna taxa in the 14 sites; the detected taxa covered about 97.9% of the assemblage’s individuals and about 99.6% of the individuals if the focus was on highly abundant taxa. In other words, the undetected proportion of soil fauna taxa was, at least, 6.9% of the total taxa; the undetected taxa covered about 2.1% of the assemblage’s individuals or about 0.4% of the individuals if only highly abundant taxa are considered. Similar interpretations can be made for temperate, subtropical and tropical regions (Figure 2 and Appendix A) as well as per sampling site (Appendix A and Appendix A). In general, the data collected in this study reflect the composition of soil fauna taxa in East Asia.

DCA analysis was conducted to show the plot ordinations (Figure 3). The highest similarity in order compositions was observed between Aershan and Changbaishan. Furthermore, Sapporo and Tahe had similar order compositions to Aershan and Changbaishan, as each was located in high-latitude temperate regions. Analogously, high similarities in order compositions were observed among the Tiantongshan, Badagongshan, Xishuangbanna and Jianfengling sites, which are all located in low-latitude subtropical and tropical regions.

The Monte Carlo permutation test showed that the total effects of latitude, MAT, MTCM, EMT and SOC reached a significant level for the overall, phytophagous and predatory orders of soil fauna composition (Table 5), and latitude, MTCM, SOC and SBD had significant effects on the composition of the saprophagous orders. The results of RDA ordination showed that the seven variables of spatial and environmental factors can explain 32.3%, 36.0%, 42.4% and 34.3% of variation in total information of soil animal composition in the overall, phytophagous, predatory and saprophagous orders, respectively (Figure 4). Specifically, the first ordination axis explains 19.4%, 20.9%, 26.4% and 21.8% of the information variation in the overall, phytophagous, predatory and saprophagous orders, respectively, and the second ordination axis explains 12.9%, 15.1%, 16% and 12.5% of the information variation, respectively. Soil fauna communities were observed to be strongly affected by spatial and environmental factors (Table 5). Community composition differed significantly in the different latitudes (R^2^ = 0.519, *p* = 0.023). Variations in fauna orders showed a strong response to soil physicochemical properties (*p* = 0.004, Figure 4a) and temperature (*p* = 0.033, Figure 4a). The dominant Lumbricida, belonging to Oligochaeta, exhibited a positive relationship with SOC or latitude and a negative relationship with SBD, MAP, MAT, MTCM and EMT values (Figure 4a, Appendix A). However, for Malacostraca, the dominant Isopoda showed a negative relationship with SOC and Latitude and a positive relationship with MAT, MTCM and EMT (Figure 4a, Appendix A). The phytophagous Diplura and Microcoryphia and the predatory Isopoda were positively correlated with SBD and negatively correlated with SOC (Figure 4b and c), whereas the predatory Geophilomorpha belonging to Chilopoda showed the opposite relationships (Figure 4c).

### 3.2. Patterns of Order Turnover

Both βj and βs increased significantly with increasing latitude difference for overall orders (βj: slope = 0.025, R^2^ = 0.057, *p* < 0.05; βs: slope = 0.019, R^2^ = 0.051, *p* < 0.05, Figure 5a; Table 6), phytophagous faunas (βj: slope = 0.035, R^2^ = 0.046, *p* < 0.05; βs: slope = 0.036, R^2^ = 0.053, *p* < 0.05, Figure 5b; Table 6) and predatory faunas (βj: slope = 0.054, R^2^ = 0.12, *p* < 0.001; βs: slope = 0.039, R^2^ = 0.12, *p* < 0.001, Figure 5c; Table 6). Both βj and βs increased significantly with the increase in EMT and SOC for overall orders (Figure 5e and Figure 6e), phytophagous (Figure 5f and Figure 6f) and predatory (Figure 5g and Figure 6g) faunas. With the increase in MAT and MTCM, both βj and βs were increased for phytophagous (Figure 6b,j) and predatory (Figure 6c,k) faunas, whereas the βj and βs of predatory faunas only were observed to be significantly increased with increasing MAP and SBD (Figure 5o and Figure 6k; Table 6). Although βj and βs were increased with increasing latitude difference and SOC, the order turnover rate of phytophagous faunas increased less than that of predatory faunas (Table 6). In addition, βs decreased significantly with the increase in MAT (slope = 0.005, R^2^ = 0.065, *p* < 0.001, Figure 6d; Table 6) and MTCM (slope = 0.059, R^2^ = 0.054, *p* < 0.001, Figure 6l; Table 6), and increased significantly with the increase in SOC and SBD (SOC: slope = 0.082, R^2^ = 0.059, *p* < 0.01, Figure 5h; SBD: slope = 0.063, R^2^ = 0.055, *p* < 0.05, Figure 5l; Table 6) for saprophagous orders.

### 3.3. Determinants of Order Turnover

Four climate factors, two soil factors and the latitude were used to assess the effects of the environment and space. The partial RDA analysis indicated that the effects of the environment and space were 54.09% and 13.84%, respectively, for overall faunas (Figure 7). The pure effects of climate on soil fauna composition were the strongest (31.27%) followed by pure soil factors (15.78%) and pure spatial factors (8.31%) for overall fauna order compositions (Figure 7a). Therefore, environmental factors (primarily climate) explained most of the variation in overall soil fauna order compositions.

For phytophagous faunas, the pure effects of climate were 29.95%, the pure effects of soil were 9.65% and the pure effects of spatial factors were 6.59% (Figure 7b). Regarding phytophagous fauna composition, the pure effects of climatic factors were stronger than those of pure soil and pure spatial factors, whereas for predatory faunas, the pure effects of climatic and soil factors were basically the same (Figure 7c). Moreover, the pure effects of climate factors on soil fauna order compositions for predacity faunas were the lowest, and the pure effects of soil and the spatially structured climatic factors were the highest (Figure 7).

## 4. Discussion

A large number of soil animal study sites have been established in different regions worldwide [66,67], and many scholars have studied soil fauna regarding their ecological function [68] and soil animal coexistence mechanisms [69,70,71]. However, most of these studies were at the regional or population scales. There is a lack of studies that analyze communities with broad ranges of taxa on large scales. Thus, integrating comparative large-scale analyses could better reveal the laws of soil fauna community distributions and taxa coexistence mechanisms.

### 4.1. Soil Fauna Community Distributions

In this study, a comparative analysis was conducted on the order composition similarities of 14 forest sites in East Asia. As expected, smaller distances between the sites tended to increase similarity in orders’ composition (Figure 3). Order richness was observed to be decreased with increasing latitude (Figure 1). This was consistent with most research findings that lower latitudes provide more available resources [7,72]. Decaëns (2010) pointed to an enhanced efficiency of mutualism under tropical climates as a possible reason for a latitudinal gradient in soil animal communities [73]. However, the results of this study found that the total richness of soil animals in temperate zones was significantly higher than that in tropical and subtropical zones (Figure 1), which was in line with the results of the non-linear shifts in soil animal community with latitude studied by Petersen and Luxton (1982) [74]. Total soil animal biomass declined from temperate ecosystems (forests and grasslands) to both arctic and tropical ecosystems and were accompanied by shifts in soil animal community composition [12]. For instance, the biomass of smaller soil animal groups (Nematoda, Collembola, Enchytraeidae and Acari) decreased in tropical ecosystems compared to temperate ecosystems [12,74]. It could be that soil animals show different temperature sensitivities, with smaller soil animals having metabolic rates that increase with increasing temperature [12]. Smaller animals need more energy from their food resources to meet the higher metabolic demands that come with higher temperatures [12].

Based on the 14 sites we studied, the patterns of order turnover increased significantly with increasing latitude differences for βj and βs of overall, phytophagous and predatory fauna orders, respectively (Figure 5). These results are consistent with those obtained in previous studies [75,76,77]. The reasons for the increase in order turnover rate with increasing latitude differences could be complicated. Among the potential reasons, latitudinal gradients in climatic tolerance and sampling effects of the species pool have been mostly reported [78,79]. The hypothesis of latitudinal gradients in climatic tolerance claims that species are more climatically tolerant in high latitudes than in low latitudes [25]. Lower climatic tolerance may further lead to narrower niche breadths in tropical than in temperate mountains, thereby decreasing the likelihood of co-occurrence of different species and increasing the taxa turnover rate [79,80].

### 4.2. Driving Forces of Soil Fauna Community Construction

Recent global syntheses of soil communities have identified contrasting environmental controls on the distribution and abundance of soil animal groups [12]. Our research shows that soil texture and temperature, especially the lowest temperature in the region, are important factors in shaping the distribution pattern of soil animals in East Asia. Similar conclusions were drawn in other studies [7,29,71,81].

The relationships of soil fauna groups at a global scale broadly follow those identified for soil fungi and bacteria [12]. At the global scale, climatic factors were found to be the main factors affecting the diversity of soil fungi in global natural ecosystems [82], and the soil organic carbon level was found to be the main factor affecting fungal diversity [83] at the regional scale (Northeast China). The changes in temperature and SOC were further expected to cause microbial community composition shifts between fungal and bacterial dominance, leading to variations in trophic transfer efficiency to their soil fauna consumers [84]. The results showed that SOC and temperature, especially the extreme minimum temperature, are environmental factors that affect the overall distribution of soil fauna in East Asia. Our study generalized these results to entire soil fauna communities and showed a dominance of small soil fauna (e.g., Nematoda, Acari and Collembola) in high latitudes with low temperature and higher SOC contents, and greater abundances of larger soil fauna (e.g., Chilopoda and Coleoptera) at mid to low latitudes in more neutral temperatures with lower SOC contents. Nevertheless, our research is inconsistent with the global results showing species distribution to be controlled primarily by pH [12]. The reason for this inconsistency may be because numerous taxa show hemispheric asymmetries in latitudinal diversity gradients: trees [85], mammals [86], termites [87], birds [88], spiders [89], ants [28] and triatomids [90]. Simultaneously, predatory soil fauna are more susceptible to environmental impacts than phytophagous fauna [91], which may lead to different distribution patterns of different functional soil animals in East Asia. This may be attributed to the relatively broad ecological niches of taxa of higher trophic levels, leading to multiple environmental requirements and adaptations [92,93].

### 4.3. Ecological Processes of Soil Fauna Community Construction

Most ecological patterns and processes in nature are scale-dependent [22,39,94], meaning their community patterns and construction mechanisms could be different under different spatial scale conditions [95] and may be simultaneously controlled by processes from multiple scales [69,96]. Locally, pure spatial variables are important for regulating taxa composition, whereas spatially structured environmental factors contribute the most at the regional scale in northeast China [22]. Our results support the hypothesis that the effect of environmental processes on order turnover of soil fauna in East Asia is more important than that of neutral processes.

The mechanisms controlling patterns of soil fauna communities in East Asia were driven by environmental processes. However, because the intensity of the interaction between predators and prey increases with a decrease in latitude, the driving patterns of different feeding soil animal communities may be different [91]. This study showed that the effect of environmental processes was more important than neutral processes for phytophagous, predatory and saprophagous fauna and that the effects of latitude differences on predatory and saprophagous fauna were minimal. The active dispersal abilities of soil animals separated by the Tullgren funnel method along the soil matrix were generally very limited [97]. This may weaken the influence of the neutral process. In contrast, soil fauna were sensitive to environmental changes [67,97]. Thus, soil fauna were more affected by environmental processes.

These results contrast with the findings obtained by Zhang et al. and Gao et al., who determined that neutral processes explained more variation in soil fauna [98,99]. This discrepancy may result partly from the use of spatial extents and species variables in the two studies. Zhang et al. and Gao et al. focused only on northeast China and on the two soil fauna groups of beetles and mites [98,99]. By comparison, our study covered 14 research samples and three temperature belts in East Asia. In particular, our taxon included almost all soil mesofauna groups. Thus, we concluded that, at a large scale, environmental processes explained more variation than neutral processes in the turnover of soil fauna.

The effects of climate as an environmental factor on phytophagous and saprophagous fauna were greater than those of soil, but the effects of climate and soil factors on predators was basically the same. Climate (mainly temperature) drives the species conversion of forest trees and shrubs in East Asia [25] and lead to changes in food resources of phytophagous fauna. Therefore, the influence of climate was observed to be greater than that of soil as environmental driving factors. For predatory animals, the biotic interaction (predation behavior) strength increases towards the equator, which may cause the expansion of the resources available to the predator, leading to the expansion of the niche [91]. The width of the niche could be restricted by more environmental factors, such as climate and soil. Therefore, climate factors and soil factors were almost equally important in the construction of soil fauna communities in East Asia.

Finally, our study used order-level identification to ensure the consistency of the soil fauna taxon in all sites in East Asia. Ponge and Salmon’s (2013) study showed that environmental filtering or species sorting culminated at the family level and were obscured by convergent evolution and co-adaptation at higher taxonomic levels [100]. High family-level diversity was commonly associated with high structural or trait diversity [14,101]. Order-level classification was adequate for the needs of our research because this study only focused on the presence or absence of information on species instead of relative abundance in species. However, biological interaction processes, such as the nutritional level, food web structure, phylogeny and functional traits, are important factors that are often ignored in community ecology studies. Therefore, future research should consider additional biological factors and distinguish the effects of neutral and environmental processes on community assembly.

## 5. Conclusions

The patterns of soil fauna order turnover increased significantly with increasing latitude and environment differences in East Asia. Our results support the hypothesis that the effect of environmental processes is more important than the effect of neutral processes on taxa turnover of soil fauna because the proportion of environmental factors explaining the turnover is significantly greater than that of spatial factors. However, the mechanisms underlying such patterns of order turnover may differ among phytophagous, predatory and saprophagous faunas. Environmental processes explained more variation than neutral processes in the turnover of phytophagous, predatory and saprophagous orders. Climatic factors in environmental processes explained more variation in the turnover of phytophagous and saprophagous faunas, and soil and environment factors equally explained the variation in the turnover of predatory soil fauna at a large scale. Therefore, phytophagous, predatory and saprophagous taxa with different functional traits and trophic levels have different community assembly processes and strategies. This contributes to the understanding, maintenance and promotion of soil biodiversity conservation and ecosystem services at large scales. 

## Figures and Tables

**Figure 1 insects-13-01103-f001:**
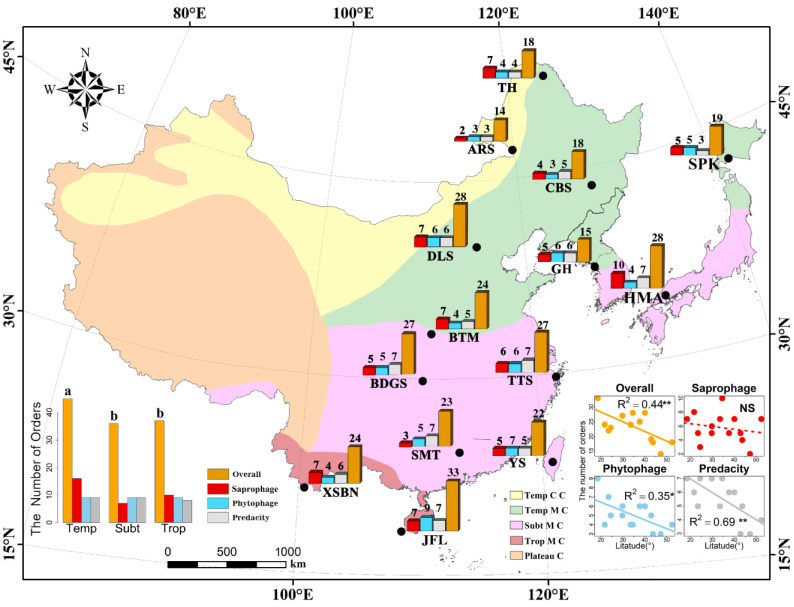
Locations of 14 sampling sites in East Asia, where TH = Tahe, ARS = Aershan, CBS = Changbaishan, DLS = Donglingshan, BTM = Baotianman, BDGS = Badagongshan, TTS = Tiantongshan, SMT = Shimentai, JFL = Jianfengling, XSBN = Xishuangbanna, GH = Guanghua, SPK = Sapporo, HMA = Hiroshima and YS = Yushan. The 14 sites belong to three climate zones, which are temperate zone (Temp), subtropical zone (Subt) and tropical zone (Trop). The graph on the bottom left shows the numbers of overall (ꭓ^2^ = 7.582, *p* = 0.02), phytophagous (ꭓ^2^ = 0.352, *p* = 0.839) and predatory (ꭓ^2^ = 1.040, *p* = 0.595) orders in temperate, subtropical and tropical zones; “a” and “b” represent significant differences. The “Tem C C”, “Temp M C”, “Subt M C”, “Trop M C” and “Plateau C” represent, respectively, temperate continental climate, temperate monsoon climate, subtropical monsoon climate, tropical monsoon climate and plateau climate. The subset of graphs on the bottom right shows the relationship between the number of orders and latitude across the 14 sampling sites. Note: “NS” indicates *p* > 0.05, “*” indicates *p* ≤ 0.05 and “**” indicates *p* ≤ 0.01.

**Figure 2 insects-13-01103-f002:**
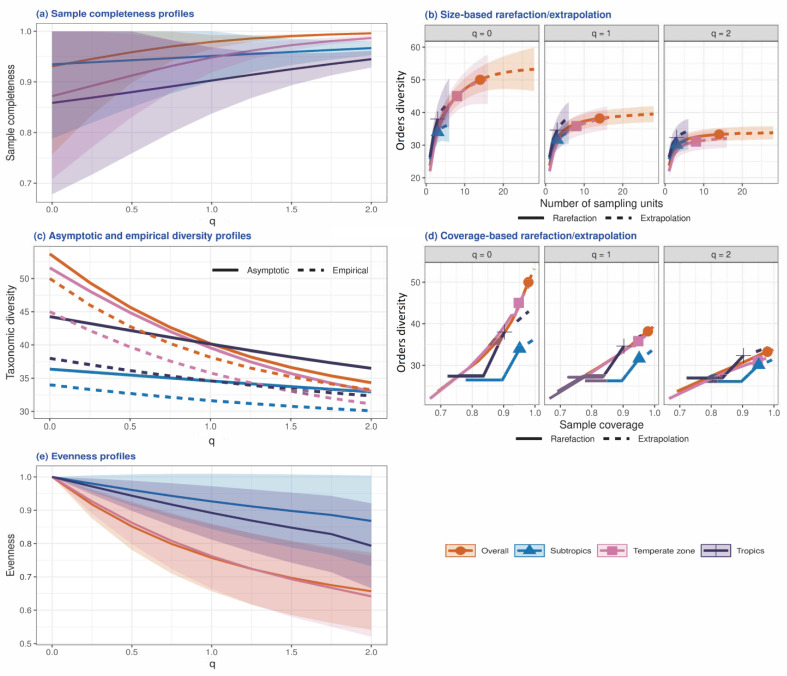
(**a**) The plots of the estimated sample completeness curves as a function of sequence q between 0 and 2 for soil fauna in East Asia. (**b**) Sample-size-based rarefaction (solid lines) and extrapolation curves (dashed lines). (**c**) The asymptotic estimates of diversity profiles (solid lines) and empirical diversity profiles (dotted lines); numerical values refer to the estimated asymptotic diversities. (**d**) Coverage-based rarefaction (solid lines) and extrapolation (dashed lines) curves up to the corresponding coverage value for a doubling of each reference sample size. (**e**) Evenness profile as a function of sequence q for 0 < q ≤ 2, based on the normalized slope of Hill numbers. Solid dots and triangles denote observed data points. All the shaded areas in (**a**–**d**) denote 95% confidence bands obtained from a bootstrap method with 100 replications. The numerical values for the three special cases of q = 0, 1 and 2 are shown in Appendix A [62].

**Figure 3 insects-13-01103-f003:**
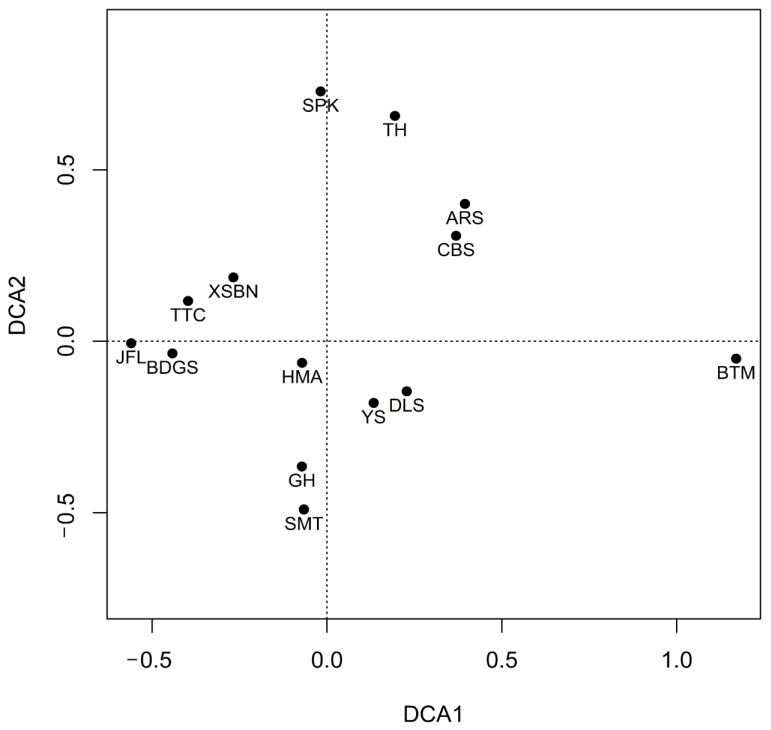
DCA analysis of soil fauna order compositions among the 14 sampling sites in East Asia, with TH = Tahe, ARS = Aershan, CBS = Changbaishan, DLS = Donglingshan, BTM = Baotianman, BDGS = Badagongshan, TTS = Tiantongshan, SMT = Shimentai, JFL = Jianfengling, XSBN = Xishuangbanna, GH = Guanghua, SPK = Sapporo, HMA = Hiroshima and YS = Yushan.

**Figure 4 insects-13-01103-f004:**
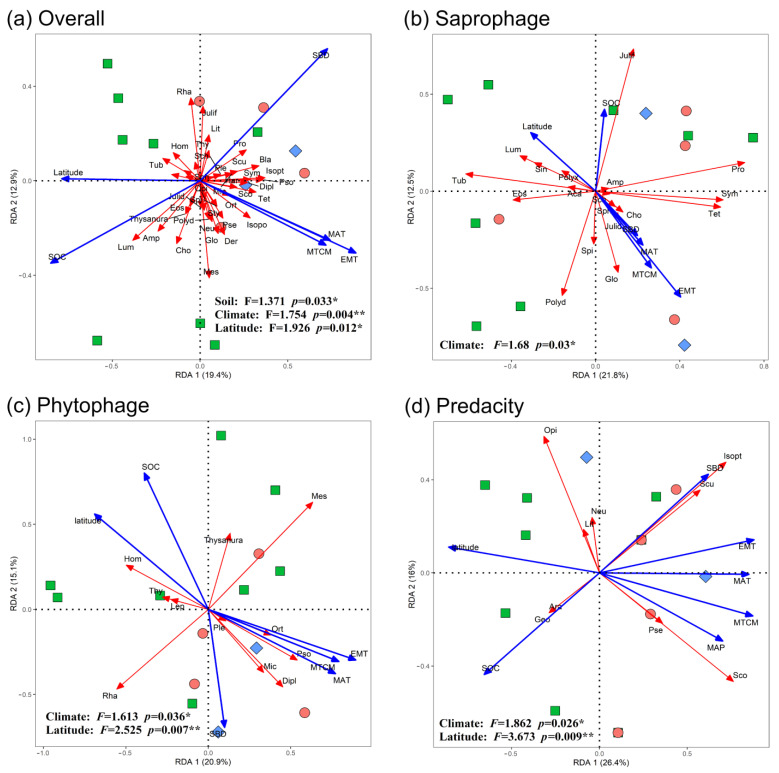
Ordination diagram for the RDA analysis for soil fauna distribution and environmental factors in East Asia. (**a**–**d**) present the overall, phytophagous, predatory and saprophagous soil fauna orders, respectively. The blue and red arrows represent environmental factors and orders, respectively. The red circles, green squares and blue diamonds represent temperate, subtropical and tropical zones, respectively. In the graphs, MAT = mean annual temperature, MTCM = mean temperature of the coldest months, EMT= extreme minimum temperature, MAP = mean annual precipitation, SOC= soil organic carbon and SBD = soil bulk density. Climate factors include MAT, MTCM, EMT and MAP. Soil factors include SOC and SBD. Order name abbreviations are shown in Table 3. Note: “*” indicates *p* ≤ 0.05 and “**” indicates *p* ≤ 0.01.

**Figure 5 insects-13-01103-f005:**
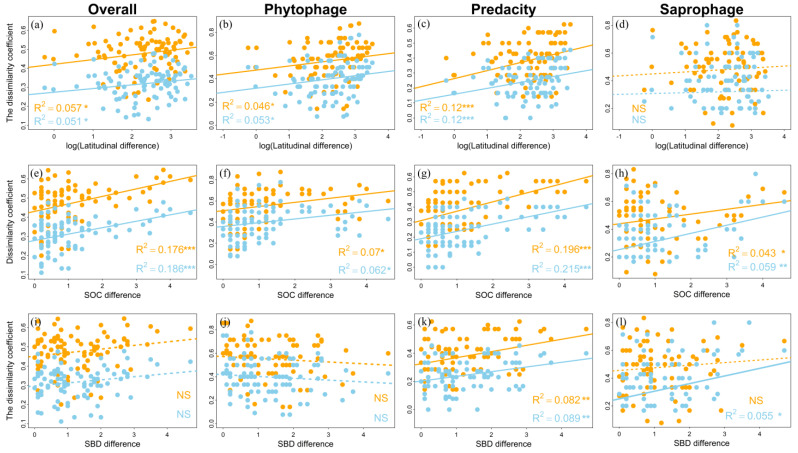
Patterns of overall (**a**,**e**,**i**); phytophagous (**b**,**f**,**j**); predatory (**c**,**g**,**k**) and saprophagous (**d**,**h**,**l**), fauna in different sampling sites along the latitude and soil factor differences in East Asia. Orange dots and lines represent βj and blue dots and lines represent βs; n = 91; “*”, “**” and “***” represent *p* ≤ 0.05, 0.01 and 0.001, respectively. “NS” indicates *p* > 0.05. (Note: SOC = soil organic carbon and SBD = soil bulk density).

**Figure 6 insects-13-01103-f006:**
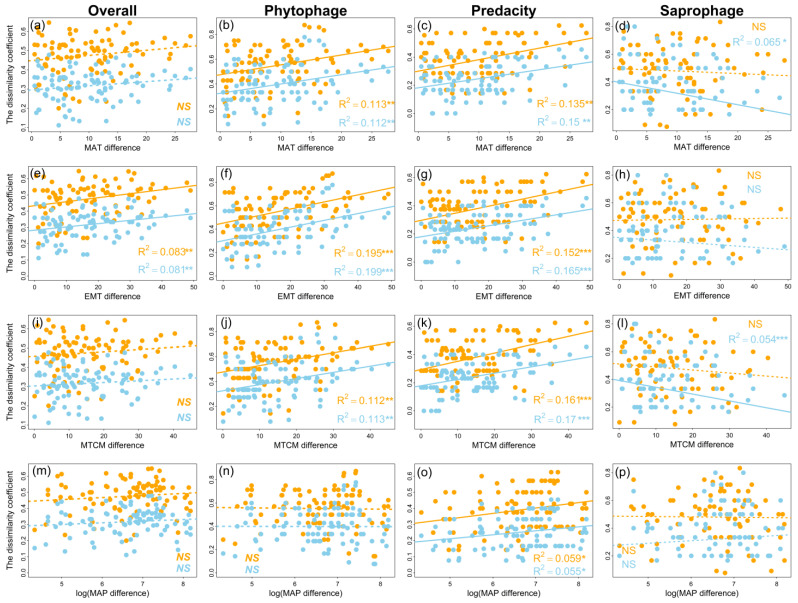
Patterns of overall (**a**,**e**,**i**,**m**); phytophagous (**b**,**f**,**j**,**n**); predatory (**c**,**g**,**k**,**o**) and saprophagous (**d**,**h**,**l**,**p**), fauna in different sampling sites with environmental differences in East Asia. Orange dots and lines represent βj and blue dots and lines represent βs; n = 91; “*”, “**” and “***” represent *p* ≤ 0.05, 0.01 and 0.001, respectively. “NS” indicates *p* > 0.05. (Note: MAT = mean annual temperature, MTCM = mean temperature of the coldest months, EMT = extreme minimum temperature and MAP = mean annual precipitation).

**Figure 7 insects-13-01103-f007:**
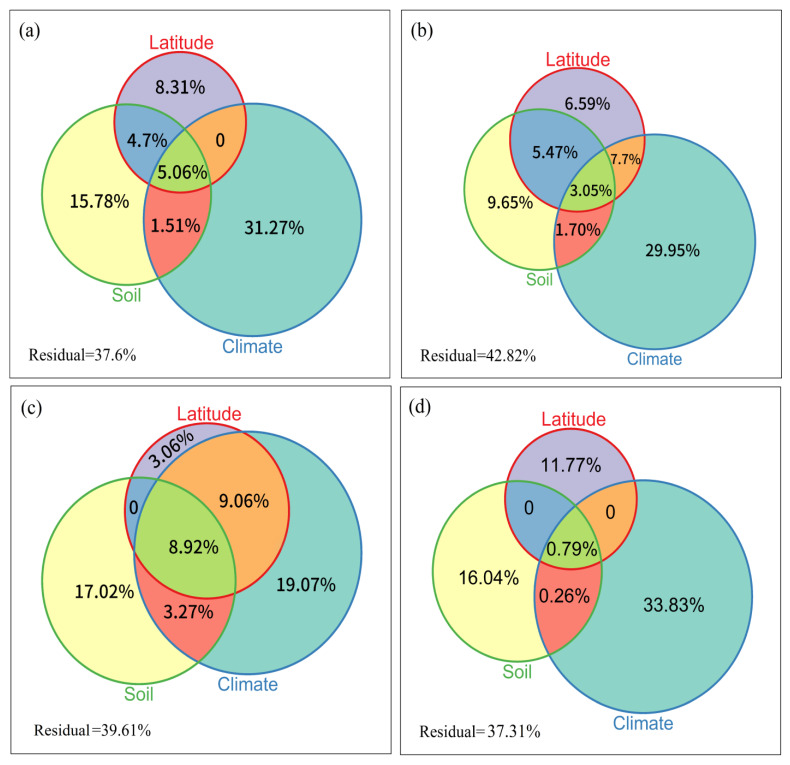
Percent of the effects of climate, soil and latitude on the overall (**a**), phytophagous (**b**), predatory (**c**) and saprophagous (**d**) fauna orders’ compositions in East Asia.

**Table 1 insects-13-01103-t001:** Summary of the soil fauna community studies included in the dataset in East Asia.

Study	Location	Vegetation	Soil Type	Climatic Region
Huang et al. [44]	Tahe	*Larix olgensis*, *Populus davidiana* and *Betula costata*	Brown coniferous forest soil	Temperate zone
	52.33° N, 124.75° E			
Zhang et al. [45]	Aershan	*Larix gmelini* and *Populus davidiana*	Brown coniferous forest soil	Temperate zone
	47.18° N, 119.94° E			
Nakamura [46]	Sapporo	*Abies fabri*	Brown forest soil	Temperate zone
	42.87° N, 141.24° E			
Han [47]	Changbaishan 43.65° N, 127.62° E	*Pinus koraiensis*, *Quercus mongolica*, *Acer mono*, *Populus davidiana* and *Betula platyphylla*	Mountain dark brown soil	Temperate zone
Xu et al. [48]	Donglingshan	*Quercus liaotungensis*	Brown soil	Temperate zone
	40.03° N, 115.47° E			
Kwon [49]	Ganghwa	*Pinus densiflora* and *Quercus mongolica*	Mountain yellow soil	Temperate zone
	37.61° N, 126.45° E			
Touyama [50]	Hiroshima	*Pinus koraiensis*, *Picea asperata* and *Tilia amurensis*	Brown forest soil	Subtropical zone
	34.52° N, 132.23° E			
Xu [51]	Baotianman 33.51° N, 111.94° E	*Quercus variabilis*, *Quercus aliena* var. *acutidentata* and *Pinus armandii*	Mountain yellow brown soil	Temperate zone
Yi [52]	Tiantongshan	*Castanopsis fargesii*, *Schima superba* and *Pinus massoniana*	Mountain yellow red soil	Subtropical zone
	29.80° N, 121.79° E			
He [53]	Badagongshan 29.74° N, 110.06° E	*Fagus lucida*, *Liquidambar formosana* and *Castanopsis fargesii*	Mountain yellow brown soil	Subtropical zone
Li et al. [54]	Shimentai 24.45° N, 113.3° E	*Castanopsis eyrei*, *Schima superba*, *Pinus massoniana* and *Pinus elliottii*	Mountainred soil	Subtropical zone
Chuan et al. [55]	Yushan	*Tsuga chinensis* and *Yushania niitakayamensis*	Mountain yellow red soil	Subtropical zone
	23.47° N, 120.89° E			
Yang et al. [56]	Xishuangbannan 21.68° N, 101.45° E	*Hevea brasiliensis*, *Mallotus paniculatus*, *Ppometia tomentosa* and *Terminalia myricocarpa*	Orthic acrisol	Tropical zone
Xiong [57]	Jianfengling 18.50° N, 109.00° E	*Antidesma maclurei*, *Vatica mangachapoi*, *Lannea grandis* and *Aporosa chinensis*	Laterite and Yellow soil	Tropical zone

**Table 2 insects-13-01103-t002:** The sampling details for 14 sites in East Asia, with TH = Tahe, ARS = Aershan, CBS = Changbaishan, DLS = Donglingshan, BTM = Baotianman, BDGS = Badagongshan, TTS = Tiantongshan, SMT = Shimentai, JFL = Jianfengling, XSBN = Xishuangbanna, GH = Guanghua, SPK = Sapporo, HMA = Hiroshima and YS = Yushan.

Site	Total Number of Individuals	Sampling Volume/L	Individuals Per m^3^	Sampling Quantity	Sampling Time/Month	Sampling Time/Year	Sampling Layer/cm	Number of Sampling Locations	Number of Repetitions
TH	12841	126	101.9	336	6, 8, 10	2004	Litter, 0–5–10–15	7	4
ARS	9684	48	201.8	128	8, 9	2004	Litter, 0–5–10–15	4	4
SPK	19012	44	432.1	440	1, 5, 7, 9	1968	0–5	11	10
CBS	24325	67.5	360.4	180	5, 7, 9	2014	Litter, 0–5–10–15	3	5
DLS	52673	270	195.1	480	4, 6, 8, 10	2012–2013	Litter, 0–5–10–15	10	4
GH	47919	39.3	1220.8	400	9	2007	0–5	20	10
HMA	18244	70.7	258.1	120	5, 7, 9	1990	Litter, 0–5	5	4
BTM	13063	47.1	277.4	640	5, 9	2018–2019	Litter, 0–5–10–15	16	5
TTS	13937	21.6	645.2	216	4, 6, 8, 10	2003–2004	0–5–10–15	6	3
BDGS	12933	52.3	247.3	360	4, 7, 11	2016	Litter, 0–15	15	4
SMT	20045	100	200.5	400	9, 10	2001	Litter, 0–5	20	5
YS	12860	32.4	396.9	324	4, 6, 8,10	1998–1999	0–5–10–15	3	3
XSBN	14434	23.6	612.9	180	7, 8, 9	1997	0–5–10–15	4	5
JFL	37083	54	686.7	540	1, 4, 7, 10	1993–1994	0–5–10–15	15	3

**Table 3 insects-13-01103-t003:** List of soil fauna orders in the 14 sites in East Asia; Sap, Omn, Phy and Pre represent saprozoic, omnivores, phytophagous and predatory, respectively.

Class	Order Name	Code	Traits	Class	Order Name	Code	Traits
Chilopoda	Geophilomorpha	Geo	Pre	Insecta	Blattaria	Bla	Omn
Chilopoda	Lithobiomorpha	Lit	Pre	Insecta	Coleoptera	Cole	Pre
Chilopoda	Scolopendromorpha	Sco	Pre	Insecta	Dermaptera	Der	Omn
Chilopoda	Scutigeromorpha	Scu	Pre	Insecta	Diplura	Dipl	Phy
Copepoda	Harpacticoida	Har	Omn	Insecta	Diptera	Dipt	Omn
Diplopoda	Chordeumatida	Cho	Sap	Insecta	Hemiptera	Hem	Omn
Diplopoda	Glomerida	Glo	Sap	Insecta	Homoptera	Hom	Phy
Diplopoda	Julida	Julid	Sap	Insecta	Hymenoptera	Hym	Omn
Diplopoda	Juliformia	Julif	Sap	Insecta	Isoptera	Isopt	Pre
Diplopoda	Polydesmida	Polyd	Sap	Insecta	Lepidoptera	Lep	Phy
Diplopoda	Polyxenida	Polyx	Sap	Insecta	Microcoryphia	Mic	Phy
Diplopoda	Sphaerotheriida	Sph	Sap	Insecta	Neuroptera	Neu	Pre
Diplopoda	Spirobolida	Spi	Sap	Insecta	Orthoptera	Ort	Phy
Pauropoda	Tetramerocerata	Tet	Sap	Insecta	Plecoptera	Ple	Phy
Gastropoda	Archaeogastropoda	Arc	Omn	Insecta	Psocoptera	Pso	Phy
Gastropoda	Mesogastropoda	Mesog	Phy	Insecta	Thysanoptera	Thy	Phy
Gastropoda	Stylommatophora	Sty	Omn	Insecta	Thysanura	Thysanura	Phy
Crustacea	Amphipoda	Amp	Sap	Oligochaeta	Lumbricida	Lum	Sap
Entognatha	Collembola	Coll	Omn	Oligochaeta	Tubificida	Tub	Sap
Entognatha	Protura	Pro	Sap	Arachnida	Acarina	Aca	Sap
Malacostraca	Isopoda	Isopo	Omn	Arachnida	Araneae	Ara	Pre
Protura	Eosentomata	Eos	Sap	Arachnida	Mesostigmata	Mes	Pre
Protura	Sinentomata	Sin	Sap	Arachnida	Opiliones	Opi	Pre
Adenophorea	Rhabditida	Rha	Phy	Arachnida	Pseudoscorpiones	Pse	Pre
Symphyla	Symphyla	Sym	Sap	Arachnida	Schizomida	Sch	Sap

**Table 4 insects-13-01103-t004:** Information on environment for 14 study sites in East Asia, where MAT = mean annual temperature, MTCM = mean temperature of the coldest months, EMT = extreme minimum temperature, MAP = mean annual precipitation, SOC = soil organic carbon and SBD = soil bulk density.

Site Name	Code	MAT	MTCM	EMT	MAP	SOC	SBD	pH
		(°C)	(°C)	(°C)	(mm)	(g/kg)	(kg/m^3^)	
Tahe	TH	−2.4	−25.5	−32.6	428.0	2.8	11.1	5.5
Aershan	ARS	−3.2	−21	−31.5	441.4	2.2	11.8	6.5
Changbaishan	CBS	3.4	−15.6	−24.8	758.0	2.6	11.2	5.7
DongLingshan	DLS	8	−7	−6.0	575.0	1.8	10.5	6.9
Baotianman	BTM	15.1	1.5	−17.0	885.6	2	11.2	5.9
Badagongshan	BDGS	11.5	0.1	−0.2	2105.4	1.2	11.9	5.9
Tiantongshan	TTS	16.2	4.2	1.1	1374.7	1.4	12.7	5.8
Shimentai	SMT	20.8	10.9	4.5	2000.0	1.6	11.4	5.1
Jianfengling	JFL	24.5	19.4	16.4	2265.8	0.4	13.9	6.2
Xishuangbanna	XSBN	21.8	10	5.00	1556.0	1	12	6
Guanghwa	GH	10.3	−8.6	−8.6	1450.5	1.2	12.9	6
Sapporo	SPK	8.5	−4	−14.1	738.0	5	9.2	5.1
Hiroshima	HMA	13.9	5	−2.0	1700.0	1.8	11.1	5.1
Yushan	YS	9	4.3	−6.00	4000.0	1.6	11	4.4

**Table 5 insects-13-01103-t005:** Monte Carlo permutation test results of environmental factors in East Asia. “*”, “**” and “***” represent *p* ≤ 0.05, 0.01 and 0.001, respectively. (Note: MAT = mean annual temperature, MTCM = mean temperature of the coldest months, EMT = extreme minimum temperature, MAP = mean annual precipitation, SOC = soil organic carbon and SBD = soil bulk density.).

	Overall		Phytophagous		Predatory		Saprophagous	
	R^2^	Pr (>r)	R^2^	Pr (>r)	R^2^	Pr (>r)	R^2^	Pr (>r)
Latitude	0.5227	0.02 *	0.6539	0.004 **	0.6086	0.01 **	0.406	0.046 *
MAT	0.4886	0.028 *	0.5362	0.016 *	0.5298	0.019 *	0.1762	0.34
MTCM	0.4714	0.035 *	0.566	0.012 *	0.6468	0.004 **	0.4775	0.019 *
EMT	0.3641	0.085	0.3293	0.098	0.5203	0.01 **	0.1626	0.35
MAP	0.7765	0.001 ***	0.7527	0.001 ***	0.7322	0.001 ***	0.3843	0.062
SOC	0.6292	0.002 **	0.5848	0.001 ***	0.3942	0.04 *	0.6665	0.002 **
SBD	0.7279	0.001 ***	0.4508	0.028 *	0.2802	0.162	0.6523	0.003 **
pH	0.0631	0.692	0.1238	0.5	0.2282	0.25	0.1134	0.35

**Table 6 insects-13-01103-t006:** The rate of dissimilarity among order composition across all possible plot pairs along the spatial or environmental gradient in East Asia. The “*”, “**”, “***” and “NS” represent *p* ≤ 0.05, 0.01, 0.001 and *p* > 0.05, respectively. (Note: Lat = latitude, MAT = mean annual temperature, MTCM = mean temperature of the coldest months, EMT = extreme minimum temperature, MAP = mean annual precipitation, SOC = soil organic carbon and SBD = soil bulk density.).

	**Overall**				**Phytophagous**				**Predatory**				**Saprophagous**			
	**βj**		**βs**		**βj**		**βs**		**βj**		**βs**		**βj**		**βs**	
	**Slope**	** *p* **	**Slope**	** *p* **	**Slope**	** *p* **	**Slope**	** *p* **	**Slope**	** *p* **	**Slope**	** *p* **	**Slope**	** *p* **	**Slope**	** *p* **
Lat	0.025	*****	0.019	*****	0.035	*****	0.036	*****	0.054	*******	0.039	*******	0.014	NS	0.006	NS
MAT	0.003	NS	0.002	NS	0.008	******	0.008	******	0.008	******	0.006	******	0.002	NS	0.010	*****
EMT	0.002	******	0.002	******	0.006	*******	0.006	*******	0.005	*******	0.004	*******	0.000	NS	0.001	NS
MTCM	0.001	NS	0.001	NS	0.005	******	0.005	******	0.006	*******	0.004	*******	0.002	NS	0.005	*****
MAP	0.013	NS	0.010	NS	0.04	NS	0.001	NS	0.034	*****	0.025	*****	0.000	NS	0.022	NS
SOC	0.039	*******	0.033	*******	0.039	*****	0.035	*****	0.062	*******	0.049	*******	0.035	*****	0.059	******
SBD	0.018	NS	0.016	NS	0.015	NS	0.016	NS	0.043	******	0.033	******	0.019	NS	0.055	*****

## Data Availability

Data is contained within the article or Appendix A. The data presented in this study are available in Table 3, Table 4 and Appendix A.

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
