# Peer review of "Environmental Effects on Taxonomic Turnover in Soil Fauna across Multiple Forest Ecosystems in East Asia"

_insects, 2022, doi:10.3390/insects13121103_

Round 1
Reviewer 1 Report (Previous Reviewer 1)
The presented paper is actual becouse the study aimed at addressing knowledge gape in the exploration of the influence of spatial end environmental factors on soil meso- macro- fauna communities. But some main aspects are either not considered or discussed.
It is important to understand that the soil fauna in its functional significance is very diverse. The trophic groups of phytophages and predators are analyzed in paper. Are where the saprophages (almost the entire mesofauna), which participate in the biotic cycle? Probably, the theory of neutrality will just explain the turnover of saprophages in latitudinal gradient as they have low dispersal abilities (line 107). Also the hypothesis of functional equivalence is the cornerstone of neutral theory. The article lacks the application of this theory to the analysis of available data. I recommend adding an analysis similar to the one carried out for saprophages.
Further comments on the text:
line 20 and 34 - add saprophages = saprozoic
line 23 and 24 - add reference (Hubbell, 2001)
line 36-38 "Our results..." is not clear.
line 59 - mistake "saporous", probably saprozoic
line 93-94 "earthworm community composition is determined by the mobility of the organisms [29]" . It is not true. Earthworms are divided into functional groups, some species inhabit deep soil layers, other species inhabit soil litter or upper soil layers. Earthworms have low mobility compared to other macrofauna. [29] reference - The microbial pan- genome - is not suitable. The article "Global distribution of earthworm diversity", 2019 is not cited.
line 99 - The article "Soil macroinvertebrate communities: a world-wide assessment", 2022 is not cited.
line 117-119 - task 2 is unclear as difference of used similarity indices is not discussed in the article. Why you can't use a single similarity index?
line 120 - It is necessary to specify the trophic groups of soil invertebrates in task 3, include saprophages.
line 183-185 - table 3 is needed, but it must be rebuilt either according to the principle of group dimension (meso- and macrofauna) or according to functional groups (Sap, Pre, Phy, Omn). In the first option it is worth classifying taxa (all millipede orders, then all insect orders, etc.)
line 451-463 - it worth refining.
line 470-471 - almost all soil mesofauna groups are saprophages! There is no analysis of this group in the article!
The article is interesting but it is needed in improvement.
Author Response
Dear Reviewer:
Thanks for the very valuable comments on our manuscript from reviewers, which have greatly helped us to further strengthen our manuscript. All suggested corrections and comments have been significant modified based on your suggestions. The whole manuscript also has been examined and modified carefully. The content modified has been marked in yellow background in the manuscript. The manuscript has undergone English language editing by MDPI. The text has been checked for correct use of grammar and common technical terms, and edited to a level suitable for reporting research in a scholarly journal. We are very grateful for your careful modification of our manuscript again.
The details are as follows:
- The trophic groups of phytophages and predators are analyzed in paper. Are where the saprophages (almost the entire mesofauna), which participate in the biotic cycle? Probably, the theory of neutrality will just explain the turnover of saprophages in latitudinal gradient as they have low dispersal abilities (line 107). Also the hypothesis of functional equivalence is the cornerstone of neutral theory. The article lacks the application of this theory to the analysis of available data. I recommend adding an analysis similar to the one carried out for saprophages.
Response: Thank you for your constructive suggestion. The original version of the manuscript did ignore the analysis of saprophage soil animals. To supplement the functional equivalence hypothesis which is the cornerstone of neutral theory. According to your suggestions, we have supplemented the analysis and detailed description of the saprophage soil animals in the manuscript. Figure 1 Figure 4, Figure 5, Figure 6, Figure 7, Table 5, and Table 6 have been added with the results of the analysis of the saprophage soil animals. (Line 147-152; Line 190; Line 252; Line255, 258; Line 292, 293, 296, 299; Line 320, 321; Line 337-351)
- line 20 and 34 - add saprophages = saprozoic
Response: Thank you for your constructive suggestion. According to your suggestions, we have added the results of saprophage soil animals. (Line 20 , 36; Page 1)
- line 23 and 24 - add reference (Hubbell, 2001)
Response: Thank you for your constructive suggestion. According to your suggestions, we have added the reference (Hubbell, 2001) in Line 113 as the ref. [38].
- line 36-38 "Our results..." is not clear.
Response: According to your suggestions, in order to avoid unclear expression, we changed "environmental process" to "environmental factor" and "neutral process" to "spatial factor". (Line 39,40; Page 1)
- line 59 - mistake "saporous", probably saprozoic
Response: Thank you for your constructive suggestion. We have changed “saporous” into “saprozoic”. (Line 62; Page 2)
- line 93-94 "earthworm community composition is determined by the mobility of the organisms [29]". It is not true. Earthworms are divided into functional groups, some species inhabit deep soil layers, other species inhabit soil litter or upper soil layers. Earthworms have low mobility compared to other macrofauna. [29] reference - The microbial pan- genome - is not suitable. The article "Global distribution of earthworm diversity", 2019 is not cited.
Response: We thank you very much for your careful review and constructive suggestion of this manuscript. It is not true that "earthworm community composition is determined by the mobility of the organisms [29]". So we changed this statement into “climate variability has a greater impact on earthworm community composition than soil properties”. And the reference changed into “29: Phillips, H.R.P. Global distribution of earthworm diversity. Science 2019, 366, 480-485.” (Line 97-78; Page 3)
- line 99 - The article "Soil macroinvertebrate communities: a world-wide assessment", 2022 is not cited.
Response: We thank you very much for your careful review and constructive suggestion of this manuscript. We have changed the reference into “Lavelle, P. Soil macroinvertebrate communities: A world-wide assessment. Glob. Ecol. Biogeogr. 2022, 21, 1261-1276.”
- line 117-119 - task 2 is unclear as difference of used similarity indices is not discussed in the article. Why you can't use a single similarity index?
Response: We simply described in the introduction "utilizing different methods to measure the biotic turnover rates can better avoid the error caused by methods of measurement." (Line 91-92 Page2)
- line 120 - It is necessary to specify the trophic groups of soil invertebrates in task 3, include saprophages.
Response: Thank you for your constructive suggestion. We have changed “functional forms of soil animals” into “saprophage, phytophage, and predacity soil animals.” to specify the trophic groups of soil invertebrates in task 3. (Line 124-125 Page 3)
- line 183-185 - table 3 is needed, but it must be rebuilt either according to the principle of group dimension (meso- and macrofauna) or according to functional groups (Sap, Pre, Phy, Omn). In the first option it is worth classifying taxa (all millipede orders, then all insect orders, etc.)
Response: According to your suggestions, we have modified Table 3 as required. We have arranged the orders according to the classification of class. (Line Page Table 3)
- line 451-463 - it worth refining.
Response: Thank you for your constructive suggestion. According to your suggestions, we have deleted “Our study area and the obtained taxon were different from those of the study of Gao et al. [22]. Therefore, our results are not entirely consistent with those of Gao’s.” (Line 465 Page 18)
- line 470-471 - almost all soil mesofauna groups are saprophages! There is no analysis of this group in the article!
Response: Thank you for your constructive suggestion. According to your suggestions, we have analyzed and described the saprovorous soil animals in the full manuscript
Reviewer 2 Report (Previous Reviewer 2)
The authors carefully responded to all comments. I am satisfied with the changes made.
Author Response
Dear Reviewer :
We are very grateful for your review of our manuscript in your busy schedule, and we also appreciate your recognition and affirmation of our research. The whole manuscript also has been carefully examined once again. The manuscript has undergone English language editing by MDPI. The text has been checked for correct use of grammar and common technical terms, and edited to a level suitable for reporting research in a scholarly journal. Thank you again for all the work you put into our manuscript.
Reviewer 3 Report (Previous Reviewer 3)
The authors have improved several aspects of their manuscript. However, there are still major and minor issues that the authors have not addressed properly and that need to be addressed if they want to achieve a level that warrants publication in an international journal:
Major issues:
- Please be careful with sticking too close to other research papers to avoid plagiarism. For example, parts of your introduction are too close to the paper by:
Chen, Y., Yuan, Z., Li, P., Cao, R., Jia, H., Ye, Y. (2016): Effects of Environment and Space on Species Turnover of Woody Plants across Multiple Forest Dynamic Plots in East Asia. Frontiers in Plant Science 7, 1533.
Please check throughout the manuscript that you avoid plagiarism.
- Please revise the abbreviations that you use for order names and standardize them throughout the manuscript. In Table 3, you use “Mes” for Mesostigmata and “Mesog” for Mesogastropoda. However, in Table S1, you use “Mesos” for Mesostigmata and “Mes” for Mesogastropoda. Please check also the associated figures for a consistent use of all order names and abbreviations!
- The use of English language still needs to be improved – in some parts of the manuscript, errors in language and style hamper the comprehension of the text. Please have your manuscript professionally checked, preferably by a native speaker.
Minor issues:
- Line 23: Change “neutral” into “neutral processes”.
- Line 24: Here, it should probably be “dispersal limitation” instead of “neutral processes” in line with the Abstract?
- Line 24: Change “reginal” into “regional”.
- Lines 50: Change “2mm” into “2 mm”.
- Lines 52: Change “2cm” into “2 cm”.
- Line 85: Change “usually based on a single indicator” into “are usually based on a single indicator”.
- Figure 1: Please depoliticize the map, as the state colors are not relevant for your study. I would suggest that you display all land areas in the same greyish color or choose colors according to the three climatic regions that you used.
- Table 1: Missing blank space in the row of “He [53]” (“Castanopsisfargesii”).
- Table 2: Change title of last column into “Number of repetitions” (instead of “Number of Repetitions”).
- Table S2: The caption title “Table S2” should be in bold consistently with the other tables.
- Lines 302-304: I cannot find these correlations described for Mesogastropoda. In Figure 4b, the “Mes” arrow appears quite orthogonal to SOC and SBD?
- Table S5: Please add a reference to this table in your manuscript.
- Table S5: Please indicate which RDA values are shown here. Do they belong to the RDA performed on all soil fauna orders (Fig. 4a)?
Table 2: Here, you have the sampling time given as months, but the years are missing.
Author Response
Dear Reviewer:
Thanks for the very valuable comments on our manuscript from reviewers, which have greatly helped us to further strengthen our manuscript. All suggested corrections and comments have been significant modified based on your suggestions. The whole manuscript also has been examined and modified carefully. The content modified has been marked in yellow background in the manuscript. The manuscript has undergone English language editing by MDPI. The text has been checked for correct use of grammar and common technical terms, and edited to a level suitable for reporting research in a scholarly journal. We are very grateful for your careful modification of our manuscript again.
The details are as follows:
- Please be careful with sticking too close to other research papers to avoid plagiarism. For example, parts of your introduction are too close to the paper by: Chen, Y., Yuan, Z., Li, P., Cao, R., Jia, H., Ye, Y. (2016): Effects of Environment and Space on Species Turnover of Woody Plants across Multiple Forest Dynamic Plots in East Asia. Frontiers in Plant Science 7, 1533.Please check throughout the manuscript that you avoid plagiarism.
Response: We thank you very much for your careful review and constructive suggestion of this manuscript. We have carefully compared and modified the introduction part to avoid plagiarism. (Line 83-90 Page 2; Line 96-97 Page 2; Line 99-103 Page 3; Line 106-108 Page 3; Line 110-112 Page 3).
Biotic turnover pattern refers to the pattern of Biotic composition divergence or biotic turnover among communities in different habitats along environmental gradients [18], and it provides considerable assistance for understanding and revealing the mecha-nism of community construction, especially the coexistence and community pattern of species at large scales [19,20]. However, in contrast to alpha diversity, patterns of bio-logical turnover have been inadequately studied [21], and measures of biotic turnover in previous studies are usually based on a single indicator [22], either Jaccard or Sørensen index or others. (Line 83-90, Page 2)
We have removed the “For example, recent studies have demonstrated that dispersal abilities were a key factor for both woody plants and birds in biotic turnover spatially” (Line 95-96 Page 2)
For instance, previous studies showed that climate variability has a greater impact on earthworm community composition than soil properties. (Line 96-97, Page 2)
Despite our accumulated knowledge about biogeographic patterns of soil biota, the underlying mechanisms of the distribution patterns remain unexplored, especially the patterns of comprehensive taxa turnover across latitudes [31].Environmental filtering and spatial dispersal of species are the main driving forc-es for changes in ecological communities and biodiversity [25] (Line 99-103, Page 3)
Several research studies indicated that biotic coexistence was attributed to differences in the biotic (e.g., community structure and species composition) and abi-otic (e.g., climate and soil) environments [33, 34, 35]. (Line 106-108, Page 3)
In addition, the influence of dispersal limitation (distance effect) on community con-struction is one of the most important corollaries of the neutral theory, which states that biotic coexistence is due to biogeographic barriers and low dispersal capacity [36,38]. (Line110-112, Page 3)
- Please revise the abbreviations that you use for order names and standardize them throughout the manuscript. In Table 3, you use “Mes” for Mesostigmata and “Mesog” for Mesogastropoda. However, in Table S1, you use “Mesos” for Mesostigmata and “Mes” for Mesogastropoda. Please check also the associated figures for a consistent use of all order names and abbreviations!
Response: We thank you very much for your careful review and constructive suggestion of this manuscript. All species order names and abbreviations have been carefully corrected. “Mesos” for Mesostigmata and “Mes” for Mesogastropoda were confirmed and consistent throughout in our manuscript. (Table3, Line187, Page 5)
- The use of English language still needs to be improved – in some parts of the manuscript, errors in language and style hamper the comprehension of the text. Please have your manuscript professionally checked, preferably by a native speaker.
Response: The manuscript has undergone English language editing by MDPI. The text has been checked for correct use of grammar and common technical terms, and edited to a level suitable for reporting research in a scholarly journal.
- Line 23: Change “neutral” into “neutral processes”.
Line 24: Here, it should probably be “dispersal limitation” instead of “neutral processes” in line with the Abstract?
Response: We thank you very much for your constructive suggestion. However, according to the suggestion of the academic editor, we have rewritten the Simple Summary section and have therefore removed the “neutral”, “neutral processes”, and “reginal”. (Line 14-28 Page 1)
- Lines 50: Change “2mm” into “2 mm”.
Lines 52: Change “2cm” into “2 cm”.
Response: We thank you very much for your careful review of this manuscript. We have changed “2mm” into “2 mm”, and changed “2cm” into “2 cm”. (Line 52, 54 Page 2)
- Line 85: Change “usually based on a single indicator” into “are usually based on a single indicator”.
Response: We thank you very much for your careful review of this manuscript. According to your suggestions, we have changed “usually based on a single indicator” into “are usually based on a single indicator”. (Line 98 Page 2)
- Figure 1: Please depoliticize the map, as the state colors are not relevant for your study. I would suggest that you display all land areas in the same greyish color or choose colors according to the three climatic regions that you used.
Response: We thank you very much for your careful review of this manuscript. According to your suggestions, we have changed the Figure 1 which display the color according to the climatic regions that we used. (Line 104 Page 4)
- Table 1: Missing blank space in the row of “He [53]” (“Castanopsisfargesii”).
Response: We thank you very much for your careful review of this manuscript. According to your suggestions, we have changed “Castanopsisfargesii” into “Castanopsis fargesii” (Table 1, Line 165).
- Table 2: Change title of last column into “Number of repetitions” (instead of “Number of Repetitions”).
Response: We thank you very much for your careful review of this manuscript. According to your suggestions, we have changed “Number of Repetitions” into “Number of repetitions” (Table 2, Line 174, Page 6).
- Table S2: The caption title “Table S2” should be in bold consistently with the other tables.
Response: We thank you very much for your careful review of this manuscript. According to your suggestions, we have changed the caption title “Table S2” in bold (Table S2).
- Lines 302-304: I cannot find these correlations described for Mesogastropoda. In Figure 4b, the “Mes” arrow appears quite orthogonal to SOC and SBD?
Response: There is indeed some ambiguity in the expression of Lines 302-304, so we have changed this part to “The phytophagous Diplura and Microcoryphia and the predatory Isopoda were positively correlated with SBD and negatively correlated with SOC (Figure 4b and c), whereas the predatory Geophilomorpha belonging to Chilopoda was observed as the opposite (Figure 4c).” (Line 308-312 Page 11)
- Table S5: Please add a reference to this table in your manuscript.
Table S5: Please indicate which RDA values are shown here. Do they belong to the RDA performed on all soil fauna orders (Fig. 4a)?
Response: The value of Table S5 belong to the RDA performed on all soil fauna orders According to your suggestions, we have changed the title of Table S5 as “The RDA values for all soil fauna orders in East Asia.”. Meanwhile, we have added the reference of Table S5 in the manuscript. (Line 306 and 308, Page10)
- Table 2: Here, you have the sampling time given as months, but the years are missing.
Response: We thank you very much for your careful review of this manuscript. According to your suggestions, we have added the “Years of Sampling time” in Table 2. (Table 2, Line 174, Page 6).
Round 2
Reviewer 1 Report (Previous Reviewer 1)
Only small clarifications:
1. Table 3 - Myriapoda consists classes Chilopoda, Diplopoda, Pauropoda. It is better to put these classes next to each other in the table.
2. Figure 7 - a, b, d: residual =, c: predacity = Is it typo?
Author Response
Dear Reviewer:
We are very grateful for your review of our manuscript in your busy schedule, and we also thank you very much for your careful review and constructive suggestion of this manuscript. The whole manuscript also has been carefully revised according to your suggestions Thank you again for all the work you put into our manuscript.
The details of the modification are as follows:
- Table 3 - Myriapoda consists classes Chilopoda, Diplopoda, Pauropoda. It is better to put these classes next to each other in the table.
We thank your constructive suggestion of this manuscript. We have modified the table 3 according to your suggestion. (Line 187 Page 6, Table3)
- Figure 7 - a, b, d: residual =, c: predacity = Is it typo?
We thank you very much for your careful review of this manuscript. We have changed the " Predacity " in Figure 7c to " Residual ". (Line 389 Page 15, Figure 7)

Reviewer 3 Report (Previous Reviewer 3)
The authors have made good efforts in revising the manuscript. The quality of the manuscript has improved much as compared to the original submission (well done!), so that it may be accepted for publication after addressing a few remaining issues:
Major issue:
- Please revise the abbreviations that you use for order names and standardize them throughout the manuscript. I am fine with a consistent use of “Mesos” for Mesostigmata and “Mes” for Mesogastropoda. However, you still use the “Mesog” in Figure 4 (a). Please check all your figures for a consistent use of all order names and abbreviations according to your study results!
Minor issues:
- Line 149: Change “Tem C” into “Temp C C” according to the legend used in the figure.
- Line 153: Change “p ≥ 0.05” into “p > 0.05” (as you have “p ≤ 0.05” as “*”).
- Figure S3: The text in the figure caption should have all the same size (at the moment it varies between 10.5 pt and 12 pt).
- Figure S3: Change “soli” into “soil” (figure caption).
- Lines 356, 366, and 372: Change “p ≥ 0.05” into “p > 0.05” (as you have “p ≤ 0.05” as “*”).
Author Response
Dear Reviewer:
We are very grateful for your review of our manuscript in your busy schedule, and we also thank you very much for your careful review and constructive suggestion of this manuscript. The whole manuscript also has been carefully revised according to your suggestions Thank you again for all the work you put into our manuscript.
The details of the modification are as follows:
- Please revise the abbreviations that you use for order names and standardize them throughout the manuscript. I am fine with a consistent use of “Mesos” for Mesostigmata and “Mes” for Mesogastropoda. However, you still use the “Mesog” in Figure 4 (a). Please check all your figures for a consistent use of all order names and abbreviations according to your study results!
We thank you very much for your careful review of this manuscript. We have changed the " Mesog " in Figure 4a to " Mes". (Line 318 Page 12, Figure 4)
- Line 149: Change “Tem C” into “Temp C C” according to the legend used in the figure.
We thank you very much for your careful review of this manuscript. We have changed the “Tem C” in Figure 1 to “Temp C C”. (Line 149 Page 4, Figure 1)
- Line 153: Change “p ≥ 0.05” into “p > 0.05” (as you have “p ≤ 0.05” as “*”).
Lines 356, 366, and 372: Change “p ≥ 0.05” into “p > 0.05” (as you have “p ≤ 0.05” as “*”)
We thank you very much for your careful review of this manuscript. We have changed the Change “p ≥ 0.05” into “p > 0.05” (Line 153 Page 4; Line 356 Page 13; Line 365 Page 14; Line 371 Page 14;)
- Figure S3: The text in the figure caption should have all the same size (at the moment it varies between 10.5 pt and 12 pt).
Figure S3: Change “soli” into “soil” (figure caption).
We thank you very much for your careful review of this manuscript. We have changed the title font of Figure S3 to the same size to achieve a unified standard. And changed “soli” into “soil” (Table S3)

This manuscript is a resubmission of an earlier submission. The following is a list of the peer review reports and author responses from that submission.
Round 1
Reviewer 1 Report
The authors made a huge statistical analysis unfortunately of the wrong data.It is not clear which orders belong to the soil micro- or mesofauna.
Moreover, it is unclear how the trophic groups were distinguished.
The composition of the soil fauna from 14 sites is unknown.
Meanwhile, there are standard methods for accounting of different groups of
soil fauna. There is no data on the number of habitats in each site.
How many sample sites were analyzed? It is not clear which sites belong to
each natural zone. What is the difference between spatial and climatic factors?
The most important fact is noted in the discussion of the results.
The authors say that their results differ from those of previous studies. Why?
Becouse it is correct to carry out such statistical analysis on any one group
of soil fauna (the authors cite researches on ants, nematodes, and earthworms
as examples). Claiming to be a global analysis of soil fauna communities, it is
necessary to apply the methods of counting of soil invertebrates according to
the ISO-standard methods.
Author Response
Thanks for the very valuable comments on our manuscript from reviewers, which have greatly helped us to further strengthen our manuscript. All suggested corrections and comments have been significant modified based on your suggestions. The whole manuscript also has been examined and modified carefully. The content modified has been marked in red in the manuscript. A major professional scientific editing company has been employed to improve the sentence of the manuscript again. We are very grateful for your careful modification of our manuscript.
The details are as follows:
- It is not clear which orders belong to the soil micro- or mesofauna.
Response: Thanks for your careful modification of our manuscript. The order information in this study was obtained from the taxon catalogs of 14 regions distributed in East Asia that have been studied. In 14 studies it has been shown that Tullgren funnels were used to isolate and capture soil fauna in their studies (Page5 Line149-151). The bottom of Tullgren funnels is sieve with 2 mm aperture, so soil animals separated and captured through the Tullgren funnels device generally have a body width of 2mm or less. Based on the body width, the type of order we studied was soil micro- or mesofauna (Page 2 Line 50-51).
- It is unclear how the trophic groups were distinguished.
Response: Thanks for your careful modification of our manuscript. The order’s trophic groups of soil faunas involved in our study are derived from the existing literature (Page 5 Line 155 ref [31, 40, 42, 43]), and have been recommended and confirmed by relevant experts.
- The composition of the soil fauna from 14 sites is unknown.
Response: Thank you for your useful suggestion. The composition of the soil fauna from 14 sites have been studied and published in relevant papers, so this part of the content is not included in this manuscript.
- There are standard methods for accounting of different groups of soil fauna.
Response: Thank you for your constructive suggestion. In 14 studies it has been shown that Tullgren funnels were used to isolate and capture soil fauna in their studies (Page5 Line149-151). And Pictorial Keys to Soil Animals of China and Catalogue of Life China were used to calibrate and confirm order lists in this study, even though the identification criteria for order lists in each study were generally consistent. (Page 5 Line 153 ref [40,42])
- There is no data on the number of habitats in each site.
Response: Thank you for your constructive suggestion. The vegetation types in the study areas include temperate coniferous and broad-leaved mixed forest temperate deciduous broad-leaved forest, temperate deciduous broad-leaved forests, subtropical ever-green broad-leaved forests, and tropical rain forests (Page 4 Line 131-133). Basically, each site is based on the primary local forest habitat. Vegetation and soil information are shown in Table1(Page 5 Line 158).
- How many sample sites were analyzed? It is not clear which sites belong to each natural zone.
Response: Thank you for your constructive suggestion. Data from 14 sampling sites were analyzed comprehensively (Page 4 Line 125,126), and each natural zone have been added to Table 1 (Page 5 Line158).
- What is the difference between spatial and climatic factors?
Response: Thank you for your constructive suggestion. Climatic factors include “mean annual precipitation”, “mean annual temperature”, “mean temperature of the coldest months”, and “extreme minimum temperature”, which belong to the temperature and humidity data. Meanwhile, Spatial factors mainly include latitude and longitude, which represent the distance in space. (Page 6 Line 175-177)
- The authors say that their results differ from those of previous studies. Why? Becouse it is correct to carry out such statistical analysis on any one group of soil fauna (the authors cite researches on ants, nematodes, and earthworms as examples).
Response: Thank you for your constructive suggestion. As noted in the manuscript, studies of the turnover rates of soil fauna at the regional scale have focused on specific species, such as earthworms, termites, skippers, nematodes, and even protozoa. However, Soil fauna contribute primarily to ecosystem functioning through trophic and/or non-trophic effects. Under the dynamic changes of predation and predation, different structures of food webs are formed. Therefore, compared to previous studies, the study of biotic turnover pattern among multiple species is of great significance for understanding the material cycle and energy flow in the ecosystem.
- Claiming to be a global analysis of soil fauna communities, it is necessary to apply the methods of counting of soil invertebrates according to the ISO-standard methods.
Response: Thank you for your constructive suggestion. The ISO-standard methods can indeed standardize soil arthropod counting, which is a considerable research standard. However, this study is a comprehensive analysis based on the data of existing studies. All the studies cited in this manuscript have used scientific counting methods. Meanwhile, the calculation of this study is based on the data of the presence/absence of species, which does not require species counting data (Page 7 Line 194,195). We will seriously consider using the ISO-standard methods as research methods in future research.
Reviewer 2 Report
The article presents extended data analysis of mesofaunal occurrence in East Asia. The Authors tried to explain space and species turnover in the forest ecosystem. The article is prepared in great detail, but in my opinion, it is difficult to understand in many parts. Please find the detailed comments below:
Simple summary: In the main, a simple summary should be simpler. For instance please do not provide numbers, like the percentages of the total variance.
Keywords: Not all keywords are necessary. For instance "neutral processes" or "Tullgren funnels" do not bring much information.
Abstract: Please explain, that the analyses were calculated based on the data of other authors.
Introduction:
Line 65: Please explain the words "alpha diversity" and "beta diversity" (line 83)
Materials and methods:
The description of this part is not adequately described.
The first impression is that the authors sampled and analyzed the mesofauna themselves. Only later do we find out that the authors use the data of other authors. Please handle this chapter in such a way that there would be no such confusion.
Figure 1. Why the results (like a number of orders) is presented in the methods part?
Table 1 and Table 2 can be combined together. If the Authors use the abbreviations of the sites, please use is consequently in the text (abbreviations in Table 2, full names in Table 1).
Table 1. The sampling area differed to a large extent between sites. Consequently differed the total number of individuals. It would be more correct if the number of individuals would be presented as the mean number of ind. or the mean number per m2. Also for further comparisons, not the row data, but means should be used. Could the Authors explain, what kind of data they used for data analysis?
Line 166: Please add the equations for Jaccard and Sørensen index. Please check the spelling of the Sørensen index in the text.
Table 4. Please consider whether or not to move Table 4 to supplementary material.
Results:
Please consider, whether or not some of the graphs should not be moved to the supplementary materials. In my opinion, some of the results are the next steps to obtaining the determinants of species turnover (for instance Figure 2, Figure 6). I leave this issue to the Authors' discretion.
Discussion: In my opinion, the discussion is too long and some parts are not necessary. For instance lines, 400-421 are the descriptions of other authors' results.
Author Response
Thanks for the very valuable comments on our manuscript from reviewers, which have greatly helped us to further strengthen our manuscript. All suggested corrections and comments have been significant modified based on your suggestions. The whole manuscript also has been examined and modified carefully. The content modified has been marked in red in the manuscript. A major professional scientific editing company has been employed to improve the sentence of the manuscript again. We are very grateful for your careful modification of our manuscript.
The details are as follows:
- Simple summary: In the main, a simple summary should be simpler. For instance please do not provide numbers, like the percentages of the total variance.
Response: Thank you for your constructive suggestion. According to your suggestions, we have removed “The environment explained 54.09, 50.62, and 57.34% of the total variance, and spatial factors explained 13.84, 15.91, and 21.04% of the total variance in species composition of overall, phytophage, and predacity faunas, respectively.” from Simple summary (Page 1, Lines 19).
- Keywords: Not all keywords are necessary. For instance "neutral processes" or "Tullgren funnels" do not bring much information.
Response: We have removed "neutral processes" or "Tullgren funnels" from Keywords (Page 1, Lines 43-44).
- Abstract: Please explain, that the analyses were calculated based on the data of other authors.
Response: We thank you very much for your careful review of this manuscript. We have explained that the analyses were calculated based on the data of other authors in our manuscript. We have added “The analyses were calculated based on published data from 14 independent sampling sites in East Asia” in our manuscript (Page 1, Lines 28-29).
- Line 65: Please explain the words "alpha diversity" and "beta diversity" (line 83)
Response: We thank you very much for your careful review of this manuscript. We have explained the words "alpha diversity" and "beta diversity" in our manuscript. We have added “The significance of community diversity studying aims at understanding the structure and function of community, and diversity indexes are often used to evaluate the status of communities or ecosystems, thus to propose corresponding conservation measures. The diversity indexes usually include α-diversity, which indicates diversity within the same community, and β-diversity, which indicates biodiversity between different communities.” in our manuscript. (Page 2, Lines 76-81)
- The first impression is that the authors sampled and analyzed the mesofauna themselves. Only later do we find out that the authors use the data of other authors. Please handle this chapter in such a way that there would be no such confusion.
Response: We explained the source of the data for this manuscript. We have explained the words “In this study, we extracted data from 14 previously studied soil fauna species catalogs located in different climate zones across East Asia, and conducted a comprehensive analysis. and the 14 samplings were:” in our manuscript. (Page 4, Lines 125-127)
- Figure 1. Why the results (like a number of orders) is presented in the methods part?
Response: We thank you very much for your careful review of this manuscript. This is a picture that contains comprehensive information. In order to include more information in the figure, we combine multiple pieces of information in the figure. However, in order to show the information of sampling location, we put this figure in the Materials and Methods as Figure 1. However, the result information of the number of species orders in Figure 1 was not expressed in the Materials and methods, but appeared in the Results.
- Table 1 and Table 2 can be combined together. If the Authors use the abbreviations of the sites, please use is consequently in the text (abbreviations in Table 2, full names in Table 1). Table 1. The sampling area differed to a large extent between sites. Consequently differed the total number of individuals. It would be more correct if the number of individuals would be presented as the mean number of ind. or the mean number per m2. Also for further comparisons, not the row data, but means should be used. Could the Authors explain, what kind of data they used for data analysis?
Response: Thank you for your constructive suggestion about Table1. According to your suggestions, we have added “Individuals per m3” in the Table 2. Meanwhile, the 14 scientific studies selected in this paper declared that the sampling completeness of each study could fully reflect the composition of local soil fauna community. Not only that, only presence/absence of species information for subsequent data processing, rather than individual richness of orders. So, presence/absence of orders was used for data analysis in our study. (Page 7, Lines 192; Page 19, Lines 489)
- Line 166: Please add the equations for Jaccard and Sørensen index. Please check the spelling of the Sørensen index in the text.
Response: We have added the the equations for Jaccard and Sørensen index in Line194-195. And we have checked the spelling of the Sørensen index in the text. (Line 29, 31, 190)
- Table 4. Please consider whether or not to move Table 4 to supplementary material.
Response: Thank you for your constructive suggestion. According to your suggestions, we have moved Table 4 to supplementary material as Table S2.
- Please consider, whether or not some of the graphs should not be moved to the supplementary materials. In my opinion, some of the results are the next steps to obtaining the determinants of species turnover (for instance Figure 2, Figure 6). I leave this issue to the Authors' discretion.
Response: Thanks very much for your pertinent suggestion about supplementary materials. We have moved some of the information in Table S1 to the text as Table 3.
- Discussion: In my opinion, the discussion is too long and some parts are not necessary. For instance lines, 400-421 are the descriptions of other authors' results.
Response: We thank you very much for your careful review of this manuscript. We have comprehensively modified the content of Line400-421 in Discussion. We have deleted “For example, global earthworm communities were observed to be strongly impacted by climatic variables, and nematode abundances were observed to be increased with an increase in SOC content. However, these soil animal communities were also found to be influenced by other additional factors. For instance, soil acidity was found to influence global earthworm communities across natural and managed ecosystems, with higher species richness at intermediate soil pH levels. Nematode abundances were found to be decreased with increasing soil pH on a global scale. And interestingly termites were influenced by climatic factors on a regional scale, with termite diversity being highest in moist lowland tropical rainforest ecosystems” (Page 17 Line421)
Reviewer 3 Report
This is an interesting study evaluating environmental effects on soil faunal communities at the taxonomic order level at study sites across East Asia. Studies on the large diversity of soil organisms, their habitat requirements and environmental effects on their distributions are still rare. Soil biodiversity includes quite different taxa of insects, which are analyzed in this study. Therefore, this study matches (in part) the scope of Insects. However, there are also myriapods, arachnids, crustaceans, annelids, nematodes, and molluscs, which are also all covered in this study. To optimize the outreach of this study among the readers of the journal Insects, I suggest that the authors (A) give a better overview on the studied soil organisms and their taxonomy and (B) elaborate on the specific role of insects in the soil, for example regarding soil processes and functions.
Furthermore, some parts of the manuscript need a major revision in order to be suitable for publication. Especially the Methods (and partly also the Results) need to be clarified and explained in more detail. During the revision, the authors should especially pay attention to the correct and consistent use of terminology. Additionally, the use of English language needs to be improved – in some parts of the manuscript, errors in language and style hamper the comprehension of the text. In the following, I provide comments on major and minor issues regarding these and further aspects.
Major issues:
- Please be careful with sticking too close to other research papers to avoid plagiarism, especially when you do not cite them. For example, parts of your introduction are too close to the paper by:
Chen, Y., Yuan, Z., Li, P., Cao, R., Jia, H., Ye, Y. (2016): Effects of Environment and Space on Species Turnover of Woody Plants across Multiple Forest Dynamic Plots in East Asia. Frontiers in Plant Science 7, 1533.
Another example is given in the presentation of your results with Table 4 and Figure 2. I understand that this may be due to the iNEXT procedure, but at least you need to cite the paper by Chao et al. (ref. [48]) in the captions of the table and figure.
Please check throughout the manuscript that you avoid plagiarism.
- In the text of the Methods section, it remains unclear if this study is only based on data from the literature (according to the studies cited in Tab. 1 and Tab. S1) or if also own field analyses were included. Starting in Chapter 2.1, you introduce study sites and refer to the Tullgren funnel, so this lets the reader think that you did these field experiments in those sites and this study is to present your results. However, in my (later) understanding, you used data from the literature that refers to those study sites (?). If this is the case, you should reformulate the Methods section, clearly stating from the beginning that it is a study based on data from the literature.
- Throughout the study, you switch between terms like “species turnover”, “species composition”, “species diversity”, “species abundances”, …. and “order turnover”, “order composition”, … It is important that you consistently refer to those taxonomic ranks that you have data on and that you use in your study. In my understanding, you analyse soil faunal communities at the taxonomic order level, so you should make a clear statement about this in the Methods section and adapt terminology throughout the manuscript. Please, be also aware of the ambiguous use of the term “order” due to different contexts. It might be confusing, for example, that you also have the order q from the iNEXT steps (see Figure 2). The different use of the term “order” should be explained. Sometimes it could also be helpful to use the term “taxa”.
- You always refer to “effects of environment and space” or similar. If I understand your analyses right, space is only considered as latitude. However, environment has also a spatial component and the latitude is of course also related to a changing environment. Therefore, I would recommend to reduce the term “effects of environment and space” (or similar) to the term “environmental effects” throughout the manuscript. In Section 2.2 you describe the environmental effects, namely geographic location (longitude and latitude – longitude is never used in your analysis?), climatic factors and soil factors. Then it should be clear, which environmental effects you analyze, and you can always just use the term “environmental effects”.
- Your study is based on the data from a bunch of references (Tab. 1, Tab. S1). To enhance the merit of your study, you should more clearly introduce the value of these studies and which aspects these studies are missing regarding the environmental effects. In the following, you should elaborate on the novelty of your study and stress the increased value achieved by combining data from those studies and environmental data (which originate from these studies or not?).
- From the overview in Table 2, it would be quite interesting if there are further patterns according to different methods, e.g. soil depths or seasons. Did you run your analyses for those data subsets?
Minor issues:
- Title: I recommend to reduce “Effects of Environment and Space” to “Environmental effects” and to change “Species Turnover” into “Taxonomic Turnover”.
- Lines 18-20: Probably, you would want to avoid those detailed numbers for the “simple summary”?
- Line 24: Change into “neutral processes”.
- Lines 30, 32, and 166: Change into “Sørensen’s”.
- Line 40: Change into “explained variation equally for turnover of predacity fauna”.
- Line 41: Change “reginal” into “regional”.
- Line 58: Check the format. Commas look different and there are extra blank spaces.
- Lines 58-61: Reformulate, especially “which is order classification as taxa” is hardly comprehensible.
- Line 66: What do you mean by “single method”? It should be “single methods” or “a single method”, but probably it would be better if you briefly explain the methods or give an example.
- Lines 74-77: Please check use of singular/plural and reformulate.
- Line 79: Change “the studies” into “studies”.
- Line 80: Add some of the “limited” literature.
- Line 81: Change into “Environmental filtering”, which is the correct term.
- Line 82: Change “significance on” into “significance for”.
- Line 88: Delete “was”.
- Lines 93-96: Your study focusses on East Asia and not on the global scale, so I suggest that you reformulate this sentence. Moreover, what do you mean by “comprehensive taxa”?
- Line 99: Give more details on the two measures of turnover.
- Line 108: Add the “N” behind the coordinates.
- Lines 112-113: This sentence should be placed in the description of your soil data (or did I understand wrong and you did your own field analyses in this study?).
- Figure 1: I would suggest that you display all countries in the same greyish color or choose colors according to the three climatic regions that you used. Moreover, change “KM” into “km” and the panel in the bottom (directly above 120°E) is not needed.
- Line 122: Change into ꭓ² (using the superscript number 2).
- Line 152: Change “latitudinal” into “latitude”.
- Lines 128-129: I would suggest du delete Table S1, as you have the citations already given in Table 1, and instead include the references in the list of references within the paper.
- Table S1: Change the title of the first column to “Authors and year”.
- Table S2: You have used the same code “Mes” for Mesostigmata and Mesogastropoda. Please check also Figure 4. Additionally, please check all the order names. For example, is it Dermaptera, Harpacticoida, and Plecoptera?
- Table 1: Please check the format, for example regarding missing blank spaces.
- Line 142: Delete “species”.
- Table 2: Here, you have the sampling time given as months, but the years are missing.
- Table 2: Change title of last column into “Number of repetitions”.
- Line 168: Delete “are”
- Lines 170-171 (Formulas 1 and 2): Check the format, for example regarding missing blank spaces.
- Line 177: Missing blank space after “SBD”.
- Table 3: Change title of the first column into “Site name”.
- Table 3: Check the format: “(kg/m³)” should go in one line.
- Table 3: Why do you have the references from Table 1 again? Are all the data from these references? You need to clarify, which of the data is from the references and which is from other sources (Lines 154-161 are too unspecific).
- Line 185: Change “analysis” into “analyzing”.
- Line 186: Which study site belongs to which climatic region?
- Line 194: I suggest to write “from the R package ‘vegan’” instead of “in the R software with ‘vegan’ package”.
- Line 195: Reformulate, especially “significance of each environmental factor and orders distribution” is hardly comprehensible.
- Line 196: Which results exactly were not significant? Do you mean the effects of pH?
- Line 199: Change into “soil fauna taxa” or “soil fauna orders”.
- Line 208: Delete the comma after “SOC”.
- Lines 209-210: Which of the variables did you transform? Probably βj and βs?
- Lines 215-217: Here you should better explain the different types of effects.
- Lines 224-228 belong to the Methods section.
- Lines 234 and 237: Change into “overall”.
- Line 246: Change into “can be made for temperate, subtropical, tropical regions (Figure 2, Table 4) and per sampling site…”.
- Table S3 and Figure S1: The value of Sapporo (SPK) does not seem to fit between the table and the figure.
- Line 279 and Lines 314-317: Change into “R²” (using the superscript number 2).
- Lines 285-287: I cannot find these correlations described for Mesogastropoda. The “Mes” arrow appears quite orthogonal to SOC and SBD?
- Figure 3: Why is it “DCA3” on the vertical axis? What about “DCA2”?
- Lines 290-294: The figure caption should be on the same page.
- Figure 4: It seems that not all taxa are presented in the figure. Which arrows are presented here? What kind of thresholds did you use?
- Line 298: Change “soli” into “soil”.
- Line 304: Figure 4 cannot be well understood without the information given in Table S2. Therefore, I suggest that you move the information from Table S2 to the Methods section of the manuscript.
- Lines 314-317 and Table 6: Do you mean “slope” instead of “slop”?
- Line 324: Here you probably mean “increased less” instead of “was less” (see Fig. 5)?
- Line 346: Change “latitude factors” into “latitude”.
- Lines 349-359: Please reformulate, it would be better to write: “pure effects of…”
- Line 355: Do you mean “phytophage” instead of “soil animal”?
- Figure 7: Here, I would also recommend to use “Latitude” instead of “Spatial” (as this is the only factor, correct?).
- Line 362: Change into “overall”, “phytophage”, and “predacity”.
- Line 365: Change into “A large number”.
- Line 368: Change “reginal” into “regional”.
- Line 368: References [56,57] are studies on global scale, so they should be cited in Line 366.
- Line 372: Missing blank space after “4.1”.
- Lines 379-381: Here it is unclear if you refer to your own study or to one of the cited ones. Where can I find this?
- Lines 457 and 460: These should be cited as “Zhang et al.” and “Gao et al.”.
- Line 465: Change “was” into “were”.
- Line 468: Change “led” into “and lead” or into “leading”.
Author Response
Thanks for the very valuable comments on our manuscript from reviewers, which have greatly helped us to further strengthen our manuscript. All suggested corrections and comments have been significant modified based on your suggestions. The whole manuscript also has been examined and modified carefully. The content modified has been marked in red in the manuscript. A major professional scientific editing company has been employed to improve the sentence of the manuscript again. We are very grateful for your careful modification of our manuscript.
The details are as follows:
Major issues:
- To optimize the outreach of this study among the readers of the journal Insects, I suggest that the authors (A) give a better overview on the studied soil organisms and their taxonomy and (B) elaborate on the specific role of insects in the soil, for example regarding soil processes and functions.
Response: Thank you for your constructive suggestion. According to your suggestions, we have added an overview of soil organisms and their taxonomy, and detailed the specific roles of insects in soil processes and functions. We have added “Soils are the most biodiverse habitats on Earth, with soil fauna accounting for about 23% of all described biodiversity globally. Based on the body width, they are usually divided into microfauna (average width less than 0.1 or 0.2 mm, such as Protozoa and Nematodes), mesofauna (average width between 0.2 and 2mm, such as Collembola and Acari), macrofauna (average larger than 2 mm, like Earthworm and Spider), and mega-fauna (average larger than 2 cm, like Mole). Soil fauna contribute primarily to ecosystem functioning through trophic and/or non-trophic effects. Most micro- and/or meso-faunal groups can regulate soil microbial processes directly by feeding on detritus and microbes, and indirectly affect soil carbon and nitrogen processes. In addition, predators such as spiders can affect soil ecosystems via the trophic cascade effects of predators. However, macrofauna such as earthworms may regulate soil microbes, and thus alter soil carbon and nitrogen processes considerably by non-trophic effects. In addition, there are some phytophagous, saporous, and omnivorous soil faunas, which are often active on the surface of soil and have high biotic richness and diversity, and also have an important impact on the soil ecosystem. The diverse feeding strategies of soil fauna and the complex non-trophic relationships establish multi-dimensional soil food webs, which plays a crucial role in the material cycle and energy flow of terrestrial ecosystems. Moreover, soil animal communities may also alter litter decomposition, nutrient mineralization, soil respiration, and plant community composition.” in our manuscript. (Page 2 Line 47-65)
- Please be careful with sticking too close to other research papers to avoid plagiarism, especially when you do not cite them. For example, parts of your introduction are too close to the paper by: Chen, Y., Yuan, Z., Li, P., Cao, R., Jia, H., Ye, Y. (2016): Effects of Environment and Space on Species Turnover of Woody Plants across Multiple Forest Dynamic Plots in East Asia. Frontiers in Plant Science 7, 1533. Another example is given in the presentation of your results with Table 4 and Figure 2. I understand that this may be due to the iNEXT procedure, but at least you need to cite the paper by Chao et al. (ref. [48]) in the captions of the table and figure. Please check throughout the manuscript that you avoid plagiarism.
Response: Thanks for the reviewer's question about the reference, we have added the reference in the appropriate position in the manuscript. (Page 10 Line 278)
- In the text of the Methods section, it remains unclear if this study is only based on data from the literature (according to the studies cited in Tab. 1 and Tab. S1) or if also own field analyses were included. Starting in Chapter 2.1, you introduce study sites and refer to the Tullgren funnel, so this lets the reader think that you did these field experiments in those sites and this study is to present your results. However, in my (later) understanding, you used data from the literature that refers to those study sites (?). If this is the case, you should reformulate the Methods section, clearly stating from the beginning that it is a study based on data from the literature.
Response: We thank you very much for your careful review of this manuscript. As for the data sources raised by the reviewers, we have explained them in the beginning of the results section of the manuscript. We have added “In this study, we extracted data from 14 previously studied soil fauna species catalogs located in different climate zones across East Asia, and conducted a comprehensive analysis. and the 14 samplings were…” in our manuscript to explain the question of the data sources. (Page 4 Line 125-126)
- Throughout the study, you switch between terms like “species turnover”, “species composition”, “species diversity”, “species abundances”, …. and “order turnover”, “order composition”, … It is important that you consistently refer to those taxonomic ranks that you have data on and that you use in your study. In my understanding, you analyse soil faunal communities at the taxonomic order level, so you should make a clear statement about this in the Methods section and adapt terminology throughout the manuscript. Please, be also aware of the ambiguous use of the term “order” due to different contexts. It might be confusing, for example, that you also have the order q from the iNEXT steps (see Figure 2). The different use of the term “order” should be explained. Sometimes it could also be helpful to use the term “taxa”.
Response: Thank you for your constructive suggestion. We made detailed and comprehensive revisions to the expressions related to "species turnover", "species composition", "species diversity" and "species abundances" in the manuscript. We have modified it by using more accurate words such as “biotic”, “order”, or “taxa”. (The modifications are marked in red).
In order to avoid ambiguity caused by "order q" in Fig. 2 (a)(b)(c), we have changed "order q" to "q" in Fig. 2 (a)(b)(c). (Page10 Line 267)
- You always refer to “effects of environment and space” or similar. If I understand your analyses right, space is only considered as latitude. However, environment has also a spatial component and the latitude is of course also related to a changing environment. Therefore, I would recommend to reduce the term “effects of environment and space” (or similar) to the term “environmental effects” throughout the manuscript. In Section 2.2 you describe the environmental effects, namely geographic location (longitude and latitude – longitude is never used in your analysis?), climatic factors and soil factors. Then it should be clear, which environmental effects your analyze, and you can always just use the term “environmental effects”.
Response: We thank you very much for your careful review of this manuscript. As suggested by the reviewer, environment has also a spatial component, so we changed the title of the manuscript from "Effects of Environment and Space" to "Effects of Environment". However, in this manuscript, environment mainly refers to climate and soil physical and chemical properties, and spatial factors include geographical spatial distance. And the purpose of this paper is to separate the contribution of community construction into environmental and spatial components. Therefore, space and environment are distinguished in this paper.
- Your study is based on the data from a bunch of references (Tab. 1, Tab. S1). To enhance the merit of your study, you should more clearly introduce the value of these studies and which aspects these studies are missing regarding the environmental effects. In the following, you should elaborate on the novelty of your study and stress the increased value achieved by combining data from those studies and environmental data (which originate from these studies or not?).
Response: Thank you for your constructive suggestion. As noted in the manuscript, studies of the turnover rates of soil fauna at the regional scale have focused on specific species, such as earthworms, termites, skippers, nematodes, and even protozoa. However, Soil fauna contribute primarily to ecosystem functioning through trophic and/or non-trophic effects. Under the dynamic changes of predation and predation, different structures of food webs are formed. Therefore, compared to previous studies, the study of biotic turnover pattern among multiple species is of great significance for understanding the material cycle and energy flow in the ecosystem. Therefore, based on the current research data and environmental data, combined with species information, this study conducted a further in-depth analysis of the community order turnover pattern at the regional scale, and obtained significant results. However, there may still be some limitations in the sampling range, seasons and soil levels in this manuscript, which will be further improved in future research.
- From the overview in Table 2, it would be quite interesting if there are further patterns according to different methods, e.g. soil depths or seasons. Did you run your analyses for those data subsets?
Response: We believe that the further analysis of the data proposed by the reviewer is a considerable suggestion. However, due to the incompleteness of the literature data in soil depth and season, no further analysis was carried out on the soil depth and season scale.
Minor issues:
- Title: I recommend to reduce “Effects of Environment and Space” to “Environmental effects” and to change “Species Turnover” into “Taxonomic Turnover”.
Response: We have changed the title of the manuscript to " Environmental Effects on Taxonomic Turnover of Soil Faunas across Multiple Forest Ecosystems in East Asia" according to the reviewer's suggestion.
- Lines 18-20: Probably, you would want to avoid those detailed numbers for the “simple summary”?
Response: According to the reviewer's suggestion, we have deleted “The environment explained 54.09, 50.62, and 57.34% of the total variance, and spatial factors explained 13.84, 15.91, and 21.04% of the total variance in species composition of overall, phytophage, and predacity faunas, respectively.” in our manuscript to avoid using detailed numbers in a “simple summary”. (Page 1 Line 19)
- Line 24: Change into “neutral processes”.
Lines 30, 32, and 166: Change into “Sørensen’s”.
Line 40: Change into “explained variation equally for turnover of predacity fauna”.
Line 41: Change “reginal” into “regional”.
Response: Based on constructive comments from reviewers. We have modified “neutral processes (Line24)”, “Sørensen’s (Line 29, 31, 190)”, “explained variation equally for turnover of predacity fauna (Line 39)”, “regional (Line 41)” in the manuscript.
- Line 58: Check the format. Commas look different and there are extra blank spaces.
Response: We thank you very much for your careful review of this manuscript. We have checked and modified the format because of the punctuation format. (Line 72)
- Lines 58-61: Reformulate, especially “which is order classification as taxa” is hardly comprehensible.
Response: We thank you very much for your careful review of this manuscript. We have changed "which is order classification as taxa" to "order taxon". (Page 2 Line 73)
- Line 66: What do you mean by “single method”? It should be “single methods” or “a single method”, but probably it would be better if you briefly explain the methods or give an example.
Response: We thank you very much for your careful review of this manuscript. We have changed " single method " to " single indicator ", and explained the methods briefly as “either Jaccard or Sørensen index.” (Page 2 Line 85)
- Lines 74-77: Please check use of singular/plural and reformulate.
Response: We have checked and modified the singular/plural in the manuscript. (Page 2 Line 95-96)
- Line 79: Change “the studies” into “studies”.
Response: We have changed “the studies” into “studies”. (Page 2 Line 98)
- Line 80: Add some of the “limited” literature.
Line 81: Change into “Environmental filtering”, which is the correct term.
Line 82: Change “significance on” into “significance for”.
Line 88: Delete “was”.
Response: We have added the reference [25] in the Line 101, changed “Environmental filtration” into “Environmental filtering” (Page 3 Line 100), Changed “significance on” into “significance for” (Page 3 Line 101), and deleted “was” (Page 3 Line 109) in the manuscript.
- Your study focusses on East Asia and not on the global scale, so I suggest that you reformulate this sentence. Moreover, what do you mean by “comprehensive taxa”?
Response: Thank you for your constructive comment about Lines 93-96. We have reformulated this sentence as “on soil animal orders composition across East Asia distributed sites by synthesizing data of soil meso-macro fauna communities.” (Page 3 Line 113-114)
- Give more details on the two measures of turnover.
Response: We have added more details “identify the patterns of orders turnover along the latitude using two measures of taxon turnover, which include Jaccard and Sørensen index,” on the two measures of turnover in the manuscript. (Page 3 Line 119)
- Line 108: Add the “N” behind the coordinates.
Response: We thank you very much for your careful review of this manuscript, The “N” has been added behind the coordinates. (Page 4 Line 130)
- Lines 112-113: This sentence should be placed in the description of your soil data (or did I understand wrong and you did your own field analyses in this study?).
Response: Thanks to the reviewers for their valuable suggestion, the sentence “Method of Tullgren funnel was used for the isolation of the soil animals and then the soil animals were identified based on morphology using the same criteria in all samples [40,41]” have been placed in “2.2 Taxa and environment variables”. (Page 5 Line 149-151)
- Figure 1: I would suggest that you display all countries in the same greyish color or choose colors according to the three climatic regions that you used. Moreover, change “KM” into “km” and the panel in the bottom (directly above 120°E) is not needed.
Response: Thank you for your constructive comment about Figure 1. The “KM” has been changed into “km” in Figure 1. however, According to China's rules, the South China Sea must be added when maps are used. So, we didn't remove this little panel. Please also get reviewer's understanding.
- Line 122: Change into ꭓ² (using the superscript number 2).
Line 152: Change “latitudinal” into “latitude”.
Response: According to the reviewer's suggestion, we have changed “ꭓ2” into “ꭓ2” (Page 4 Line 141), “latitudinal” into “latitude” (Page 4 Line 144).
- Lines 128-129: I would suggest du delete Table S1, as you have the citations already given in Table 1, and instead include the references in the list of references within the paper.
Table S1: Change the title of the first column to “Authors and year”.
Table 1: Please check the format, for example regarding missing blank spaces.
Response: Thank you for your constructive comment about Table 1. According to the suggestions, we have deleted Table S1. At the same time, the references in Table1 have been placed in the main text. At the same time, wo have checked the format, for example regarding missing blank spaces. (Page 5 Line 158)
- Table S2: You have used the same code “Mes” for Mesostigmata and Mesogastropoda. Please check also Figure 4. Additionally, please check all the order names. For example, is it Dermaptera, Harpacticoida, and Plecoptera?
Response: Thanks for your careful modification of our manuscript. The order names in Table S2 have been carefully examined and modified. The code “Mes” for “Mesostigmata”, the “Mesog” for “Mesogastropoda”. Meanwhile, according to the reviewer's suggestion, we have moved the information from Table S2 to the Methods section of the manuscript as Table 3. (Page 6 Line 183)
- Line 142: Delete “species”.
Response: We have changed the “species” into “taxa” in Line 165.
- Table 2: Change title of last column into “Number of repetitions”.
Response: According to the reviewer's suggestion in Table 2. We have changed title of last column into “Number of repetitions”. (Page 6 Line 171-174), we also added the “Individuals per m3” in Table 2.
- Line 168: Delete “are”
Lines 170-171 (Formulas 1 and 2): Check the format, for example regarding missing blank spaces.
Line 177: Missing blank space after “SBD”.
Response: According to the reviewer's suggestion, we have deleted “are” in Line 192, checked the format in Line 194-195, blank space has been added after “SBD” in Line 201.
- Table 3: Change title of the first column into “Site name”.
Table 3: Check the format: “(kg/m³)” should go in one line.
Table 3: Why do you have the references from Table 1 again? Are all the data from these references? You need to clarify, which of the data is from the references and which is from other sources (Lines 154-161 are too unspecific).
Response: We are grateful to the reviewers for their careful review of the manuscript. According to the reviewer's suggestion in Table 3. “Site name” and “(kg/m³)” have been modified as required. At the same time, specific sources of data have been explained “Longitude and latitude, and soil factors were obtained from literature, and the climate factors were obtained from National Earth System Science Data Center (http://www.geodata.cn/), Google Earth Engine (earthengine.google.com/)”. (Page 6 Line 179-182)
- Line 185: Change “analysis” into “analyzing”.
Response: According to the reviewer's suggestion, “analyzing” has been changed by “analysis” (Page 8 Line 210)
- Line 186: Which study site belongs to which climatic region?
Response: We have added information about the climate zones of the study sites in the Table1. (Page 5 Line 158)
- Line 194: I suggest to write “from the R package ‘vegan’” instead of “in the R software with ‘vegan’ package”.
Line 195: Reformulate, especially “significance of each environmental factor and orders distribution” is hardly comprehensible.
Response: We are grateful to the reviewers for your careful review of the manuscript. We changed “The “envfit” function in the R software with “vegan” package was used to test the significance of each environmental factor and orders distribution” to “The “envfit” function from the R package “vegan” was used to test the significance of each environmental factor for the orders distribution” in the manuscript. (Page 8 Line 219)
- Line 196: Which results exactly were not significant? Do you mean the effects of pH?
Response: The results of Monte Carlo permutation tests in Table 5. The pH value was not significant. However, we have removed the ambiguous phrase " The pH value results were not significant in overall, phytophage, and predacity species composition, so the pH value was excluded in the subsequent analysis." (Page 8 Line 220)
- Line 199: Change into “soil fauna taxa” or “soil fauna orders”.
Response: We are grateful to the reviewers for your careful review of the manuscript. we have changed the “soil fauna in East Asia” in to “soil fauna orders in East Asia” at the title of Table 4, meanwhile, the Table 4 has moved to the supplementary materials as Table S2.
- Line 208: Delete the comma after “SOC”.
Response: We are grateful to the reviewers for your careful review of the manuscript. According to the reviewer's suggestion, we have deleted the comma after “SOC”. (Page 9 Line 229)
- Lines 209-210: Which of the variables did you transform? Probably βj and βs?
Response: In the data analysis of the manuscript, latitude and MAP data were transformed. (Figure 5 Page 14 Line348)
- Lines 224-228 belong to the Methods section.
Response: We are grateful to the reviewers for your careful review of the manuscript. We have deleted “We classified all sites into three climate types based on their geographic locations, thermal characteristics, and water availability, as follows: temperate, subtropical, and tropical. The vegetation types include temperate coniferous and broad-leaved mixed forest, temperate deciduous broad-leaved forest, subtropical evergreen broad-leaved forest, and tropical rain forest (Figure 1). Line244-248”.
- Lines 234 and 237: Change into “overall”.
Line 279 and Lines 314-317: Change into “R²” (using the superscript number 2).”
Response: According to the reviewer's suggestion, we have changed the “Overall” into “overall”. (Page 9 Line 251,254), “R2” into “R2” (Page11 Line 296)
- Line 246: Change into “can be made for temperate, subtropical, tropical regions (Figure 2, Table 4) and per sampling site…”.
Response: According to the reviewer's suggestion, we have changed original sentence into “can be made for temperate, subtropical, tropical regions (Figure 2, Table S2) and per sampling site”
- Lines 285-287: I cannot find these correlations described for Mesogastropoda. The “Mes” arrow appears quite orthogonal to SOC and SBD?
Figure 4: It seems that not all taxa are presented in the figure. Which arrows are presented here? What kind of thresholds did you use?
Response: We are grateful to the reviewers for your careful review of the manuscript. For some orders, the RDA1 and RDA2 values are too small and the arrows are too short to show in the Figure 4, so I put the results of the RDA in the supplementary materials as tableS5.
- Figure 3: Why is it “DCA3” on the vertical axis? What about “DCA2”?
Response: We have changed the name of the axes into "DCA2" in the Figure 3. (Page 11 Line 306)
- Lines 290-294: The figure caption should be on the same page.
Response: We thank you very much for your careful review of this manuscript, we have adjusted the image to ensure that the figure caption is on the same page.
- Line 298: Change “soli” into “soil”.
Response: We are very grateful for your careful modification of our manuscript. We have changed “soli” into “soil”. (Page 12 Line 315)
- Figure 4 cannot be well understood without the information given in Table S2. Therefore, I suggest that you move the information from Table S2 to the Methods section of the manuscript.
Response: According to the reviewer's constructive suggestion, we have moved the main information Table S1 to the Methods section of the manuscript as Table 3.
- Lines 314-317 and Table 6: Do you mean “slope” instead of “slop”?
Line 324: Here you probably mean “increased less” instead of “was less” (see Fig. 5)?
Line 346: Change “latitude factors” into “latitude”.
Lines 349-359: Please reformulate, it would be better to write: “pure effects of…”
Line 355: Do you mean “phytophage” instead of “soil animal”?
Response: We are very grateful for your careful modification of our manuscript. We have changed “slop” into “slope” (Table 6, Page13, Line 347); “was less” into “increased less” (Page 13 Line 341); “latitude factors” into “latitude” (Page 15 Line 362); “the effects of pure” into “pure effects of…” (Page 15 Line 369-374); “soil animal” into “phytophage faunas” (Page 15 Line 371).
- Figure 7: Here, I would also recommend to use “Latitude” instead of “Spatial” (as this is the only factor, correct?).
Line 362: Change into “overall”, “phytophage”, and “predacity”.
Response: We are very grateful for your careful modification of Figure 7, according to the reviewer's constructive suggestion, we have changed “Spatial” into “Latitude” at the Figure 7. At the same time, we have also changed “Overall”, “Phytophage”, and “Predacity” into “overall”, “phytophage”, and “predacity”, respectively; “spatial processes” into “latitude” at the title of Figure 7 (Page 16 Line 377, 378)
- Line 365: Change into “A large number”.
Line 368: Change “reginal” into “regional”.
Response: According to the reviewer's constructive suggestion, we have changed “Large number of” into “A large number of” (Page 16 Line 380); “reginal” into “regional” (Page 17 Line 383)
- Line 368: References [56,57] are studies on global scale, so they should be cited in Line 366.
Line 372: Missing blank space after “4.1”.
Response: According to the reviewer's constructive suggestion, we have moved the references [70,71] to Line 382; and added a blank space after “4.1” (Page 17 Line 387).
- Lines 379-381: Here it is unclear if you refer to your own study or to one of the cited ones. Where can I find this?
Response: In order to clarify the source of the result “However, the results of this study found that the total richness of soil animals in temperate zones was significantly higher than that in tropical and subtropical zones,” we add “Figure1” to the sentence. (Page 17 Line 396)
- Lines 457 and 460: These should be cited as “Zhang et al.” and “Gao et al.”.
Line 465: Change “was” into “were”.
Line 468: Change “led” into “and lead” or into “leading”.
Response: We are very grateful for your careful modification of our manuscript. According to the reviewer's suggestion, we have changed the cites into “Zhang et al. and Gao et al.” (Page 18 Line 467); “was” into “were” (Page 18 Line 472); “led” into “and lead” (Page 18 Line 475).